# Transferring Fairness under Distribution Shifts via Fair Consistency Regularization

**Bang An**
Department of Computer Science
University of Maryland, College Park
bangan@umd.edu

**Zora Che**
Department of Computer Science
Boston University
zche@bu.edu

**Mucong Ding**
Department of Computer Science
University of Maryland, College Park
mcding@umd.edu

**Furong Huang**
Department of Computer Science
University of Maryland, College Park
furongh@umd.edu

## Abstract

The increasing reliance on ML models in high-stakes tasks has raised a major concern about fairness violations. Although there has been a surge of work that improves algorithmic fairness, most are under the assumption of an identical training and test distribution. In many real-world applications, however, such an assumption is often violated as previously trained fair models are often deployed in a different environment, and the fairness of such models has been observed to collapse. In this paper, we study how to transfer model fairness under distribution shifts, a widespread issue in practice. We conduct a fine-grained analysis of how the fair model is affected under different types of distribution shifts and find that domain shifts are more challenging than subpopulation shifts. Inspired by the success of self-training in transferring accuracy under domain shifts, we derive a sufficient condition for transferring group fairness. Guided by it, we propose a practical algorithm with fair consistency regularization as the key component. A synthetic dataset benchmark, which covers diverse types of distribution shifts, is deployed for experimental verification of the theoretical findings. Experiments on synthetic and real datasets, including image and tabular data, demonstrate that our approach effectively transfers fairness and accuracy under various types of distribution shifts[1].

## 1 Introduction

Machine learning's social impact has broadened as it is widely used to aid decision-making in real-world applications, such as hiring, loan approval, facial recognition, and criminal justice. To avoid discrimination against a subset of the population (e.g., w.r.t race or gender), many efforts on algorithmic fairness have been carried out [12, 21, 65, 44, 46, 15, 7]. Although existing work has achieved remarkable success in ensuring fairness, most of them assume the distribution of data at test time is identical to that in the training set. However, recent studies show that the fairness of a model is likely to collapse when encountering a distribution shift. For example, [19] observes that a fair income predictor trained with data from one state might not be fair when used in other states. [50] tries to maintain fairness in healthcare settings, but a model that performs fairly according to the metric evaluated in "Hospital A" shows unfairness when applied to "Hospital B". Such observations

---

[1]Code is available at https://github.com/umd-huang-lab/transfer-fairness.

36th Conference on Neural Information Processing Systems (NeurIPS 2022).

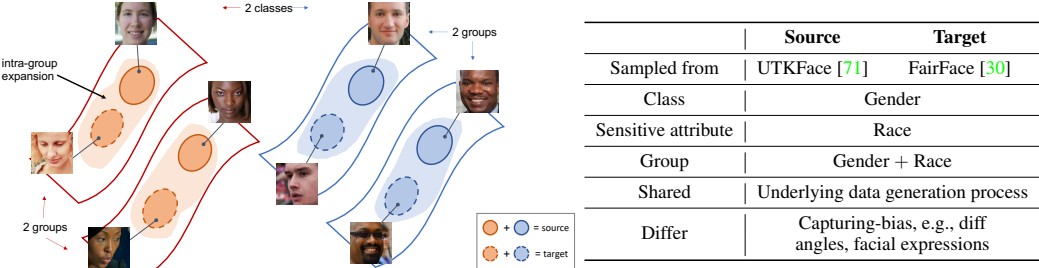

| | Source | Target |
|---|---|---|
| Sampled from | UTKFace [71] | FairFace [30] |
| Class | | Gender |
| Sensitive attribute | | Race |
| Group | | Gender + Race |
| Shared | | Underlying data generation process |
| Differ | | Capturing-bias, e.g., diff angles, facial expressions |

Figure 1: **Illustration of intra-group expansion assumption in the input space.** An example of gender classification task with the sensitive attribute being race. Intra-group expansion assumes that different groups are separated but every group is self-connected under certain transformations. If a model has consistent predictions under those transformations, we can propagate labels within each group. Under this assumption, we propose to obtain fairness and accuracy in both domains by a self-training algorithm with fair consistency regularization.

motivate us to find the reason behind the collapse of fairness and investigate how to transfer fairness under distribution shifts. Specifically, when we have labeled data in the source domain and unlabeled data in the target domain, we investigate how to adapt the fair source model to a target domain with the goal of achieving both accuracy and fairness in both domains.

Intuitively, the fairness of a model in the target domain strongly depends on the nature of distribution shifts. In this paper, we only consider cases where the oracle model is the same in two domains. We characterize distribution shifts by assuming two domains share the same underlying data generation process where data is generated from a set of latent factors with a fixed generative model, and the shift is caused by the shift of the marginal distribution of some factors. We categorize distribution shifts into three types [32]: 1) *Domain shift* where source and target distributions comprise data from related but distinct domains (e.g., train a model in hospital A but test it in hospital B). 2) *Subpopulation shift* where two domains overlap, but relative proportions of subpopulations differ (e.g., the proportion of female candidates increases at test time). 3) *Hybrid shift* where domain shift and subpopulation shift happen at the same time. We find domain shift more challenging for transferring fairness since the model's performance is unpredictable in unseen domains. Such a finding is supported empirically on a synthetic dataset that is developed to simulate diverse types of distribution shifts. While recent work explores methods to transfer fairness [54, 48, 23], most considered settings fall into subpopulation shifts. In this paper, we consider all three types of distribution shifts. Our analysis suggests we encourage consistent fairness under different factor values.

We draw inspiration from recent progress on self-training in transferring accuracy under domain shifts [61, 5, 70, 3, 49, 55]. The success of self-training is due to an *expansion assumption* and a *consistency regularization* algorithm. The expansion assumption also assumes two domains share one underlying generative model and the support of the distribution on each class is a connected compact set under data transformations (i.e., has a good continuity). Under the *expansion assumption*, [61] and [5] prove that self-training, which enforces consistent predictions for the same input under different transformations (i.e., under shifts of nuisance factors), can propagate labels from the source to the target domain. This approach exhibits superior performance in transferring accuracy [70, 49], but does not consider fairness.

Taking demography into consideration, we relax the expansion assumption to a more realistic *intra-group expansion assumption*, as shown in Figure 1, which only requires continuity of the underlying distribution within every group (i.e., data points with the same class and sensitive attribute) rather than the entire class. Based on the intra-group expansion assumption, we derive a sufficient condition that guarantees fairness in both source and target domains. This sufficient condition suggests that ensuring the trained model gains the same consistency across groups under a fair teacher classifier guarantees fairness in both domains. However, such a teacher classifier is not available in practice, and we need a practical treatment.

Guided by the theoretical algorithm, we propose a practical self-training algorithm to minimize and balance consistency loss across groups. Our algorithm builds upon Laftr [42], an adversarial learning method for fairness, and FixMatch [55], a self-training framework. To encourage similar consistency in different groups, we propose a novel *fair consistency regularization*. By reweighting the consistency loss of each group dynamically according to the model's performance, the algorithm

encourages the model to pay more attention to the high-error group while training. Our method results in a model that is fair in source and has similar consistency across groups. As indicated by our theory, it would have similar accuracy across groups in the target domain so that we can transfer fairness. We evaluate our method under different types of distribution shifts with the synthetic and real datasets. Experiments show that our approach achieves high accuracy and fairness in the target domain without sacrificing performance in the source domain. To the best of our knowledge, this is the first work using self-training to transfer fairness under distribution shifts.

**Summary of contributions: (1)** We provide a fine-grained analysis of fairness under distribution shifts and develop a synthetic dataset to study model fairness under different types of distribution shifts. **(2)** Theoretically, we derive a sufficient condition for transferring fairness under distribution shifts. **(3)** Algorithmically, we propose a theory-guided algorithm for transferring fairness with a fair consistency regularization as the key component. **(4)** Experimentally, we evaluate our method on synthetic data, real image data, and real tabular data. All results show the effectiveness of our approach in transferring fairness.

## 2 Preliminaries and Notations

**Transfer Fairness.** Let $X, A, Y$ and $\mathcal{X}, \mathcal{A}, \mathcal{Y}$ denote random variables and sample space of input features, sensitive attribute, and label. For simplicity, we assume binary sensitive attribute and binary classification, while our method can easily extend to multi-sensitive attributes and multi-class cases (see Appendix E). We aim to learn a classifier $g : \mathcal{X} \to \mathcal{Y}$ and are interested in its fairness under distribution shifts. Specifically, with $S$ and $T$ denoting source and target domains, we study how to transfer fairness and accuracy when $\mathbb{P}_S(X, A, Y) \neq \mathbb{P}_T(X, A, Y)$, with the access to $X, A, Y$ in the source domain, but only $X, A$ in the target domain. In the self-training algorithm, we use $g_{tc}$ to denote a teacher classifier, and $g^*$ to denote the oracle classifier. We use the word "group" to denote the set of data that has the same label and sensitive attribute.

**Fairness Metric.** Since we consider classification problems in this paper, we expect the fairness metrics could encourage models to achieve similar classification performance across groups. We use two metrics in this paper, *equalized odds* and *variance of group accuracy*. *Equalized odds* [27] is a widely used unfairness metric in classification problems that requires the true positive rate and the true negative rate to be the same among groups. It is defined as $\Delta_{odds} = \frac{1}{2} \sum_{y=0}^{1} \left| \mathbb{P}(\hat{Y} = y | A = 0, Y = y) - \mathbb{P}(\hat{Y} = y | A = 1, Y = y) \right|$, where $\hat{Y} = g(X)$ is the prediction. Additionally, we also evaluate the *variance of group accuracy* which is defined as $V_{acc} = Var(\{\mathbb{P}(\hat{Y} = y | A = a, Y = y), \forall a, y\})$. Smaller $V_{acc}$ indicates the model is fairer since it performs similarly across groups. Note that the variance of group accuracy can help avoid trivial fairness where a model with constant output has $\Delta_{odds} = 0$, but such fairness is meaningless.

## 3 Fairness under Distribution Shifts

In this section, we provide a fine-grained analysis of fairness under various types of distribution shifts based on a unified framework of distribution shift characterization.

**A Unified Framework to Characterize Distribution Shifts.** Following [62], we characterize distribution shifts by assuming a unified latent variable model for the underlying data generation process. We denote the underlying factors as $Y^1, Y^2, ..., Y^K$, and data point as $X$. Two of the factors are label $Y^l$ (i.e. $Y$) and sensitive attribute $Y^a$ (i.e. $A$). We call other factors *nuisance factors* since they are irrelevant to the classification task.

**Assumption 1.** *(Underlying data generation process) We assume the data is generated from a latent generative model as $\boldsymbol{y}^{1:K} \sim \mathbb{P}(Y^{1:K})$ and $\boldsymbol{x} \sim \mathbb{P}(X | Y^{1:K} = \boldsymbol{y}^{1:K})$. The generative model is fixed $\mathbb{P}_S(X | Y^{1:K} = \boldsymbol{y}^{1:K}) = \mathbb{P}_T(X | Y^{1:K} = \boldsymbol{y}^{1:K})$ but the marginal distribution of factors varies in two domains $\mathbb{P}_S(Y^{1:K}) \neq \mathbb{P}_T(Y^{1:K})$, causing the distribution shift $\mathbb{P}_S(Y^{1:K}, X) \neq \mathbb{P}_T(Y^{1:K}, X)$.*

It is realistic to assume two domains share the same data generation process. For example, the underlying physical process of cell imaging is fixed, while the distribution of underlying factors (e.g. *gender*, *age* or *equipment*) may vary in two hospitals (i.e. two domains), resulting in the distribution shift of the observed tissue images. Based on the unified framework, we consider two major types

of distribution shifts, namely *subpopulation shift* and *domain shift*, which are widely considered in many practical applications [32].

**Definition 3.1.** *(Subpopulation shift) We say it is a subpopulation shift, if for any factor $Y^i$, the sample space of it remains the same in two domains (i.e., $\mathcal{Y}_S^i = \mathcal{Y}_T^i$), but the marginal distribution of at least one factor changes (e.g., $\mathbb{P}_S(Y^j) \neq \mathbb{P}_T(Y^j)$), resulting in $\mathbb{P}_S(Y^{1:K}) \neq \mathbb{P}_T(Y^{1:K})$ and $\mathbb{P}_S(Y^{1:K}, X) \neq \mathbb{P}_T(Y^{1:K}, X)$.*

**Definition 3.2.** *(Domain shift) We say it is a domain shift, if at least one nuisance factor $Y^i, i \neq l, i \neq a$, has different sample space in two domains, $\exists y^i \in \mathcal{Y}_T^i$, but $y^i \notin \mathcal{Y}_S^i$, resulting in $\mathbb{P}_S(Y^{1:K}) \neq \mathbb{P}_T(Y^{1:K})$ and $\mathbb{P}_S(Y^{1:K}, X) \neq \mathbb{P}_T(Y^{1:K}, X)$.*

Intuitively, under subpopulation shift, the sample space overlaps, and only the marginal distributions of factors vary in the two domains. For example, the proportion of females versus males in training and deployment time differs. In contrast, under domain shift, the source model has never seen the data with factor values that only exist in the target domain. For instance, the source model is unaware of the equipment used for cell imaging at deployment time.

**Why do distribution shifts cause unfairness?** Suppose the marginal distributions of a binary nuisance factor $Y^i$ differ in two domains with $\mathbb{P}_S(Y^i) \neq \mathbb{P}_T(Y^i)$. The unfairness in two domains are

$$\Delta_{odds}^S = \mathbb{P}_S(Y^i = 0) \times \Delta_{odds}^S|_{Y^i=0} + \mathbb{P}_S(Y^i = 1) \times \Delta_{odds}^S|_{Y^i=1} \qquad (1)$$
$$\Delta_{odds}^T = \mathbb{P}_T(Y^i = 0) \times \Delta_{odds}^T|_{Y^i=0} + \mathbb{P}_T(Y^i = 1) \times \Delta_{odds}^T|_{Y^i=1}.$$

Due to the same generation process where $\mathbb{P}_S(X|Y^i = y^i) = \mathbb{P}_T(X|Y^i = y^i)$, we have $\Delta_{odds}^S|_{Y^i=y^i} = \Delta_{odds}^T|_{Y^i=y^i}, \forall y^i \in \{0, 1\}$. Under subpopulation shift, $Y^i$ has the same sample space in two domains but with different proportions (e.g., $\mathbb{P}_S(Y^i = 0) = 0.9, \mathbb{P}_S(Y^i = 1) = 0.1, \mathbb{P}_T(Y^i = 0) = 0.1, \mathbb{P}_T(Y^i = 1) = 0.9$), while under domain shift the sample space differs (e.g. $\mathbb{P}_S(Y^i = 0) = 1, \mathbb{P}_T(Y^i = 1) = 1$). It is easy to see from (1) that if a model is perfectly fair on data with $Y^i = 0$ but unfair on data with $Y^i = 1$, then the model is highly fair in the source domain but highly unfair in the target domain under both cases. Therefore, if the model has inconsistent performance on data generated from different nuisance factor values, then the shifted marginal distribution of those factors may cause fairness collapse.

**How to transfer fairness under distribution shifts?** Based on the above analysis, one way is to train the model to be fair under any values of factors. It is possible under subpopulation shift as stated in the following proposition (see proof and discussion in Appendix B).

**Proposition 3.1.** *(Transfer fairness under subpopulation shift) Consider the subpopulation shift that is caused by the shifted marginal distribution of nuisance factor $Y^i$ (i.e., $\mathbb{P}_S(Y^i) \neq \mathbb{P}_T(Y^i)$), while $\mathcal{Y}_S^i = \mathcal{Y}_T^i = \mathcal{Y}^i$. If model $f$ is strictly fair in source domain under any value of factor $Y^i$ satisfying $\mathbb{P}_S(g(X) = y^l|Y^a = 0, Y^l = y^l, Y^i = y^i) = \mathbb{P}_S(g(X) = y^l|Y^a = 1, Y^l = y^l, Y^i = y^i), \forall y^i \in \mathcal{Y}^i, y^l \in \{0, 1\}$, then model $g$ is also fair in target domain with $\Delta_{odds} = 0$.*

Our empirical results (Figure 3) also support this finding. However, domain shift is more challenging. The source model's performance on target data is unpredictable due to the distinct sample space. One promising way to tackle domain shift is to enforce the model's invariance to nuisance factors so that the source model would have the same behavior on target data. Note that this solution also works for subpopulation shift since it leads to the case in Proposition 3.1 directly. The above analysis motivates us to transfer fairness by encouraging consistent fairness under different nuisance factor values.

## 4 Transfer Fairness via Fair Consistency Regularization

### 4.1 Theoretical Analysis: A Sufficient Condition for Transferring Fairness

In reality, distribution shifts are usually hybrid, and we may not know all the underlying factor values. In this section, we consider a general case where we only have access to input $X$, label $Y$, and sensitive attribute $A$. We use data transformations to simulate the shift of nuisance factors. Our theory is based on [61] and [5] which prove that encouraging consistency under transformations can propagate labels so that to transfer accuracy. In this section, we find that in order to transfer fairness, we need a fair label propagation process that requires the model to have similar consistency across groups. We introduce assumptions and our findings as follows.

**Assumption 2** (Separability of the input). *Let $S_a^y$ and $T_a^y$ denote the sample space of $X|_{A=a,Y=y}$ in source and target domains. The ground truth class and sensitive attribute for $\boldsymbol{x} \in S_a^y \cup T_a^y$ are consistent, which are $y \in \{0,1\}$ and $a \in \{0,1\}$. We assume the sample spaces of $X$ in two domains are $S = \cup_y \cup_a S_a^y$ and $T = \cup_y \cup_a T_a^y$, where groups are separated with 1) $S_a^y \cap S_{a'}^y = T_a^y \cap T_{a'}^y = S_a^y \cap T_{a'}^y = \emptyset, \forall y, a \neq a'$, and 2) $S_a^y \cap S_{a'}^{y'} = T_a^y \cap T_{a'}^{y'} = S_a^y \cap T_{a'}^{y'} = \emptyset, \forall a, a', y \neq y'$.*

This is a realistic assumption as illustrated in Figure 1 where the data from two domains are from the same underlying conditional distribution $X|_{Y,A}$, and groups are separated by label and sensitive attribute. We define $U_a^y = \frac{1}{2}(S_a^y + T_a^y)$ as the group distribution, and $U$ as the population distribution on the entire data. Next, we characterize the good continuity of group distributions with the definition of *neighbor* and *intra-group expansion* assumption.

**Definition 4.1** (Neighbor). *Let $\mathcal{T}$ denote a set of input transformations and define the transformation set of $\boldsymbol{x}$ as $\mathcal{B}(\boldsymbol{x}) \triangleq \{\boldsymbol{x}'|\exists t \in \mathcal{T}, \text{s.t. } \|\boldsymbol{x}' - t(\boldsymbol{x})\| \leq r\}$. For any $\boldsymbol{x} \in S_a^y \cup T_a^y$, we define the neighbor of $\boldsymbol{x}$ as $\mathcal{N}(\boldsymbol{x}) := (S_a^y \cup T_a^y) \cap \{\boldsymbol{x}'|\mathcal{B}(\boldsymbol{x}) \cap \mathcal{B}(\boldsymbol{x}') \neq \emptyset\}$ and define the neighbor of a set $V \in \mathcal{X}$ as $\mathcal{N}(V) := \cup_{\boldsymbol{x} \in V \cap (\cup_y \cup_a S_a^y \cup T_a^y)} \mathcal{N}(x)$.*

Intuitively, two examples are neighbors if they are near each other after applying some transformations. Note that we only consider neighbors that have the same class and sensitive attribute (i.e., from the same group). Based on this definition, we characterize the continuity of group distribution with *intra-group expansion* assumption where any small set has a large neighbor in its group.

**Assumption 3** (Intra-group expansion). *We say that $U_a^y$ satisfies $(\alpha, c)$-multiplicative expansion for some constant $\alpha \in (0,1)$ and $c > 1$, if for all $V \subset U_a^y$ with $\mathbb{P}_{U_a^y}(V) \leq \alpha$, the following holds:*
$$\mathbb{P}_{U_a^y}(\mathcal{N}(V)) \geq \min\{c\mathbb{P}_{U_a^y}(V), 1\}.$$

Different from the *expansion* assumption proposed in [61] which considers the class continuity, *intra-group expansion* assumes group continuity. As shown in Figure 1, this is more realistic since groups are separated by both label and sensitive attribute. We can also interpret it as the transformations that change the value of nuisance factors will generate neighbors within the same group.

This assumption allows us to propagate labels within the group from one domain to another by encouraging consistency under transformations. We use $R_{U_a^y}(g) \triangleq \mathbb{P}_{U_a^y}[\exists \boldsymbol{x}' \in \mathcal{B}(\boldsymbol{x}), \text{s.t. } g(\boldsymbol{x}) \neq g(\boldsymbol{x}')]$ to denote the *consistency loss* of classifier $g$ on the group distribution $U_a^y$, which is the fraction of examples where $g$ is not robust to input transformations. Since we only have partial supervision (i.e., no labels in the target domain), we use a self-training framework to obtain a model that is accurate and fair in both domains (i.e., on $U_a^y$). Based on the theory of self-training in [61], we derive a sufficient condition in Theorem 4.1 that bounds the unfairness and error on the population distribution. We use 0-1 loss to evaluate the *error* of $g$ as $\varepsilon_{U_a^y}(g) \triangleq \mathbb{P}_{U_a^y}[g(\boldsymbol{x}) \neq g^*(\boldsymbol{x}')]$, and the *disagreement* between $g$ and a teacher classifier $g_{tc}$ as $L_{U_a^y}(g, g_{tc}) \triangleq \mathbb{P}_{U_a^y}[g(\boldsymbol{x}) \neq g_{tc}(\boldsymbol{x}')]$.

**Theorem 4.1** (Guarantee fairness). *Suppose we have a teacher classifier $g_{tc}$ with bounded unfairness such that $|\varepsilon_{U_a^y}(g_{tc}) - \varepsilon_{U_{a'}^{y'}}(g_{tc})| \leq \gamma, \forall a, a' \in \mathcal{A}$ and $y, y' \in \mathcal{Y}$. We assume intra-group expansion where $U_a^y$ satisfies $(\bar{\alpha}, \bar{c})$-multiplicative expansion and $\varepsilon_{U_a^y}(g_{tc}) \leq \bar{\alpha} < 1/3$ and $\bar{c} > 3, \forall a, y$. We define $c \triangleq \min\{1/\bar{\alpha}, \bar{c}\}$, and set $\mu \leq \varepsilon_{U_a^y}(g_{tc}), \forall a, y$. If we train our classifier with the algorithm*
$$\min_{g \in G} \max_{a,y} R_{U_a^y}(g), \qquad \text{s.t.} \quad L_{U_a^y}(g, g_{tc}) \leq \mu \quad \forall a, y$$
*then the error and unfairness of the optimal solution $\hat{g}$ on the distribution $U$ are bounded with*

$$\varepsilon(\hat{g}) \leq \frac{2}{c-1}\varepsilon_U(g_{tc}) + \frac{2c}{c-1}R_U(\hat{g}), \tag{2}$$

$$\Delta_{odds}(\hat{g}) \leq \frac{2}{c-1}(\gamma + \mu + c\max_{a,y} R_{U_a^y}(\hat{g})) \tag{3}$$

**Remark.** This sufficient condition suggests we fit a teacher classifier which is fair on the population distribution and minimize the *consistency loss* in every group. The unfairness of the resulting model is bounded by the quality (unfairness and error) of the teacher classifier and the worst-group consistency loss. Intuitively, we can understand the consistency loss as the model invariance to the nuisance factors. With a group-balanced consistency loss, the model would have similar invariance to the nuisance factors resulting in similar group performance on the unseen data so that to transfer accuracy and fairness. We also bound the variance of group accuracy with the variance of consistency loss (Appendix C). Both bounds suggest we balance and minimize the consistency loss across groups.

## 4.2 Practical Algorithm: Fair Consistency Regularization

There are two challenges in realizing the theoretical algorithm in Theorem 4.1. First, we need a high-quality teacher model, but the model trained with labeled source data is only fair and accurate in the source domain. Second, existing consistency regularization methods do not consider fairness. We tackle the first problem by leveraging the iterative self-training paradigm that updates the teacher model with the student model while training, thus making it fairer and fairer. We tackle the second problem by proposing a novel fair consistency regularization.

**Algorithm.** Figure 2 shows the overall training diagram. There are three major components:
**(1)** In every training epoch, we use the student model obtained in the last epoch as the teacher model and automatically fit the teacher model by initializing the student model to be the same as the teacher model. In other words, only one model is training itself iteratively.
**(2)** To ensure the accuracy and fairness in the source domain, we adopt Laftr

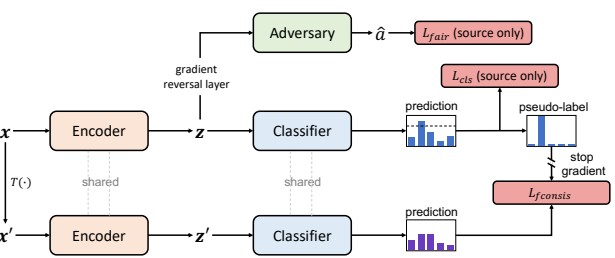

Figure 2: Training diagram.

[42], an adversarial learning method consisting of a classification loss $L_{cls}$ and a fairness loss $L_{fair}$. **(3)** To transfer fairness and accuracy, we do consistency training on all unlabeled data (including source and target data). Following FixMatch [55], we use the pseudo-labels generated by the teacher model as supervision for consistency training where the model should have consistent predictions under transformations. Different from FixMatch, we propose a fair consistency regularization with a balanced group consistency loss $L_{fconsis}$.
We train the model with the weighted summation of these three losses as shown in Figure 2. We defer the detailed loss functions of $L_{cls}$ and $L_{fair}$ with a detailed algorithm description to Appendix D.

**Fair Consistency Regularization.** To tighten the upper bound of the unfairness in Theorem 4.1, we need to minimize and balance consistency loss across groups. However, the consistency regularization in FixMatch [55] does not distinguish groups and might amplify the bias as observed in [76] and our experiments. Instead, we propose to use a fair consistency regularization that evaluates the consistency loss per group and minimizes the balanced consistency loss $L_{fconsis}$ defined as below.

$$L_{fconsis}(g) = \sum_{y=0}^{1} \sum_{a=0}^{1} \lambda_a^y L_a^y(g) \tag{4}$$

$$\text{where} \quad L_a^y(g) = \frac{1}{\sum_{\boldsymbol{x}_a^y} \mathbb{1}} \sum_{\boldsymbol{x}_a^y} \mathbb{1}(\max(g_{tc}(\boldsymbol{x}_a^y)) \geq \tau) H(\operatorname{argmax}(g_{tc}(\boldsymbol{x}_a^y)), g(t(\boldsymbol{x}_a^y))) \tag{5}$$

where $\boldsymbol{x}_a^y$ denotes an input with sensitive attribute $A = a$ and class $Y = y$. $L_a^y(g)$ is model $g$'s consistency in the group of $\{\boldsymbol{x}_a^y\}$, and $\lambda_a^y$ is the corresponding weight of the group consistency loss. Here, we abuse the usage of $g(\boldsymbol{x})$ to denote the output logits of model $g$ on input $\boldsymbol{x}$ and thus, $\operatorname{argmax}(g_{tc}(\boldsymbol{x}_a^i))$ is the pseudolabel generated by teacher classifier. $t(\boldsymbol{x}_a^y)$ is the transformed input as defined in Definition 4.1. We use a cross-entropy loss $H(\cdot)$ to encourage the consistency under transformation $t(\cdot)$ and only consider examples that the teacher model has high confidence in with a confidence threshold $\tau$. Note that data is classified into groups according to the true sensitive attribute and pseudolabels. To balance the group consistency loss, we propose to weigh each group inversely with the number of confident pseudolabels, and set $\lambda_a^y$ as

$$\hat{\lambda}_a^y = \frac{1}{\sum_{\boldsymbol{x}_a^y} \mathbb{1}(\max(g_{tc}(\boldsymbol{x}_a^y)) \geq \tau)}, \quad \lambda_a^y = \hat{\lambda}_a^y / \sum_{a,y} \hat{\lambda}_a^y. \tag{6}$$

The weights will dynamically change while training. Heuristically, if the teacher model is only confident in a few examples in a group, the model's consistency in this group is more likely to be low. With the proposed weights, a larger penalty will be applied to such groups. Therefore, the proposed fair consistency regularization will enforce the model to pay more attention to high-error groups. By doing so, the trained model would enjoy similar consistency loss across groups. Together with the self-training algorithm, it would have similar accuracy across groups in the target domain.

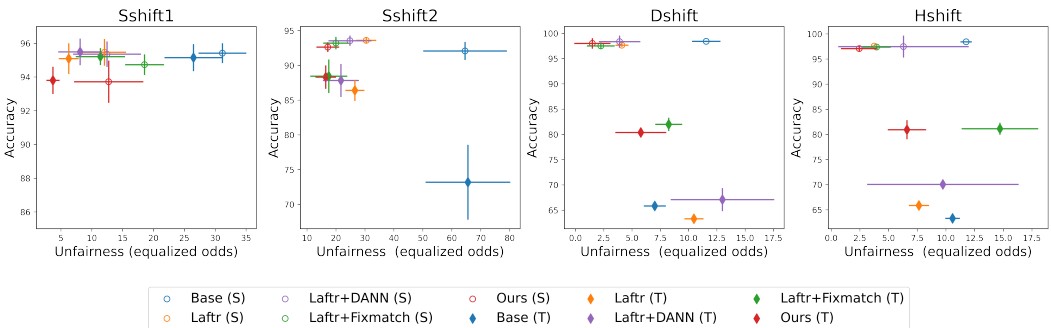

Figure 3: Accuracy and unfairness (error bar denotes the standard deviation) in two domains under subpopulation shifts (Sshift 1, Sshift 2), domain shift (Dshift), and hybrid shift (Hshif). (S) and (T) denotes the evaluation in the source and target domains respectively. Results show that domain shift is more challenging than subpopulation shift, and our method can effectively transfer accuracy and fairness under all the distribution shifts considered.

## 5   Related Work

This section features related work for transferring fairness. Another discussion of related work in fair machine learning, domain adaptation, and self-training is deferred to Appendix A. Out-of-distribution fairness remains an under-explored area. We categorize prior works into five classes. 1) *Group-wise distribution matching*. [51] derives an upper bound for fairness in the target domain which suggests training a fair model in the source domain and matching the distributions of relevant groups from two domains in feature space at the same time. [64] also applies group-wise distribution matching but with Wasserstein distance. Such methods are hard to achieve if we do not have supervision in the target domain and it also shares the drawback of distribution matching methods. 2) *Reweighting*. When the proportions of groups differ in two domains, reweighting the examples in the source domain can approximate the target distribution. [16] uses reweighting to deal with fairness problems under covariate shift and [23] uses reweighting together with a fairness test to guarantee fairness under demographic shift. Reweighting methods strongly rely on the support cover assumption which is not satisfied under domain shift. 3) *Distributionally robust optimization (DRO)*. This line of work considers unknown target data that can be any arbitrary weighted combinations of the source dataset and train a fair model that is robust to the worst-case shift [48, 43]. These methods also assume subpopulation shift instead of domain shift. 4) *Causal inference*. [54] conducts causal domain adaptation and DRO based on a well-characterized causal graph that describes the data construction and distribution shift. Causal methods highly rely on the correct causal graph which is hard to obtain in reality. For example, [50] finds that the causal graph in real applications (e.g. predicting the skin condition in dermatology) is far more complicated which violates normal assumptions, thus making those approaches inapplicable. 5) *Others*. [10] derives bound for fairness violation under distribution shifts. There are also studies that aim to maintain fairness under distribution shifts through online learning [69], and loss curvature matching [59]. To the best of our knowledge, this is the first work that uses self-training to transfer fairness. Some work also studies self-supervised learning and fairness, yet they use unlabeled data and self-training to improve the in-distribution fairness [14, 68, 9] which is different from our goal.

## 6   Experiments

### 6.1   Evaluation under Different Types of Distribution Shifts with a Synthetic Dataset

In order to study the fairness under distribution shifts and verify our theoretical findings, we develop a synthetic dataset to simulate different types of distribution shifts.

**Synthetic dataset.** The synthetic dataset is adapted from the 3dshapes dataset [31] which contains images of 3D objects generated from six independent latent factors (*shape*, *object hue*, *scale*, *orientation*, *floor hue*, *wall hue*). This dataset satisfies our assumption on the shared underlying data generation process. We simulate different types of distribution shifts by varying the marginal distributions of the latent factors and sample the data accordingly (see Appendix D.1 for details).

**Distribution shifts.** We set the image as input $X$, and select three latent factors to be class ($Y = shape$), sensitive attribute ($A = object\ hue$), and a nuisance factor that might shift ($D = scale$). We consider four widely observed distribution shifts in reality ($\mathbb{P}_S(X, Y, A, D) \neq \mathbb{P}_T(X, Y, A, D)$):
(1) **Sshift 1**: Subpopulation shift where only the nuisance factor shift (i.e. more small objects in source but more large objects in target), $\mathbb{P}_S(Y, A) = \mathbb{P}_T(Y, A)$, $\mathbb{P}_S(D) \neq \mathbb{P}_T(D)$.
(2) **Sshift 2**: Subpopulation shift where $A$ and $Y$ have different correlations in two domains (i.e. most red objects are cubes in source but are capsules in target), $\mathbb{P}_S(Y, A) \neq \mathbb{P}_T(Y, A)$, $\mathbb{P}_S(D) = \mathbb{P}_T(D)$.
(3) **Dshift**: Domain shift where the nuisance factor has different sample spaces (i.e. only small objects in source but only large objects in target), $\mathbb{P}_S(Y, A) = \mathbb{P}_T(Y, A)$, $\mathbb{P}_S(D) \neq \mathbb{P}_T(D)$, $\mathcal{Y}_S^d \neq \mathcal{Y}_T^d$.
(4) **Hshift**: Hybrid shift of (2) and (3).

**Baselines.** We do shape classification task with an MLP model and compare our method with four baselines: Base (standard ERM); Laftr; Laftr+DANN (a combination of Laftr and a domain adaptation method [22]); Laftr+FixMatch. In our method, we also use Laftr and FixMatch but with the proposed fair consistency regularization. Since the shifted nuisance factor is *scale*, we use random padding and cropping as transformations in our method and Laftr+FixMatch. We train Base and Laftr with labeled source data and train others with unlabeled target data as well.

**Domain shift is more challenging than subpopulation shift.** Figure 3 shows that under subpopulation shifts, the fair source model trained with Laftr also has high accuracy and fairness in the target domain although it has not seen any target date. This is because the sample space is shared (e.g. small and large objects both exist in the source data), and the model has similar performance under all factor values. Thus, good performance remains even if the proportion of data changes, verified Proposition 3.1. In contrast, under domain shift and hybrid shift, the fair source model performs poorly in the target domain where data is sampled from a different sample space, suggesting the difficulty of domain shift.

**Our method can transfer fairness and accuracy under various types of distribution shifts.** Under domain shift, the domain adaptation method DANN does not help in transferring fairness or accuracy. Consistency regularization forces the model to behave consistently under cropping and padding, resulting in a model that has similar predictions regardless of the object's scale and thus transfers accuracy. However, it may cause bias as shown in the results of Laftr+FixMatch. With the proposed fair consistency regularization, the model gains similar consistency across groups, resulting in a similar accuracy in all groups in the target domain and thus transfers fairness. Therefore, our method achieves high accuracy and fairness in two domains under all the considered distribution shifts.

| | Source | | | Target | | |
|---|---|---|---|---|---|---|
| | Acc | Unfairness | | Acc | Unfairness | |
| Method | | $V_{acc}$ | $\Delta_{odds}$ | | $V_{acc}$ | $\Delta_{odds}$ |
| Base | 92.85±0.49 | 2.30±0.97 | 4.81±0.69 | 74.49±0.83 | 5.79±3.49 | 9.90±1.27 |
| Laftr | 93.24±0.41 | 1.19±0.46 | 2.44±0.51 | 74.35±1.46 | 6.92±0.72 | 9.79±1.54 |
| CFair | 92.51±0.22 | 1.76±0.53 | 4.75±0.85 | 73.53±0.89 | 7.51±0.73 | 7.26±1.95 |
| Laftr+DANN | 91.33±0.08 | 2.12±1.72 | 2.70±0.67 | 74.28±1.63 | 6.25±2.59 | 8.27±2.11 |
| CFair+DANN | 90.89±0.76 | 2.01±0.70 | 4.43±1.36 | 74.62±1.06 | 6.23±0.90 | 5.26±2.07 |
| Laftr+FixMatch | 96.62±0.06 | 0.77±0.21 | 2.23±0.44 | 83.87±0.48 | 8.21±0.67 | 9.32±1.01 |
| CFair+FixMatch | 96.13±0.53 | 1.28±0.53 | 2.78±0.74 | 83.11±0.49 | 7.87±1.86 | 7.89±0.40 |
| Ours (w/ Laftr) | 96.08±0.07 | 0.96±0.39 | 2.59±0.35 | 85.52±0.40 | 2.82±0.87 | 5.70±0.52 |
| Ours (w/ CFair) | 95.65±0.22 | 1.56±0.37 | 3.85±0.97 | 84.48±0.42 | 2.88±0.99 | 5.43±0.65 |

Table 1: Transfer fairness and accuracy from UTKFace to FairFace

## 6.2 Evaluation on Real Datasets

**Evaluation on images.** We use UTKFace [71] as the source data and FairFace [30] as the target data. Although both are facial images, there is a distribution shift between them due to different image sources. We consider a gender classification task with race as the sensitive attribute. We use VGG16 [53] as the model and RandAugment [18] (excluding transformations that may change the group) as the transformation function. Additional to previous baselines, we also use CFair [73] as the method for in-distribution fairness. As shown in Table 1, there is indeed a distribution shift as the source model trained with Laftr or CFair is no longer accurate or fair in the target domain. The domain adaptation method has a limited effect on transferring accuracy and fairness. As expected, self-training (Laftr+Fixmatch and CFair+Fixmatch) significantly improves the accuracy in the target domain, but the unfairness is high. With the proposed fair consistency regularization, our method

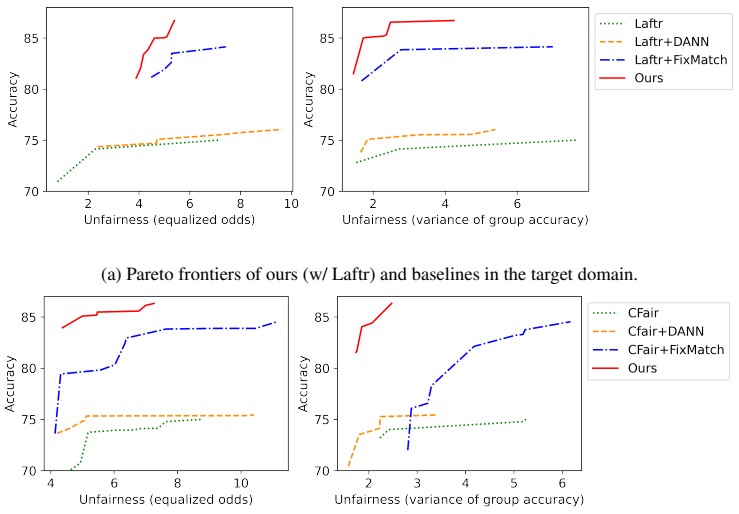

(a) Pareto frontiers of ours (w/ Laftr) and baselines in the target domain.

(b) Pareto frontiers of ours (w/ Cfair) and baselines in the target domain.

Figure 4: Comparison of Pareto frontiers. Upper left is preferred. Our method outperforms baseline methods in achieving accuracy and fairness at the same time.

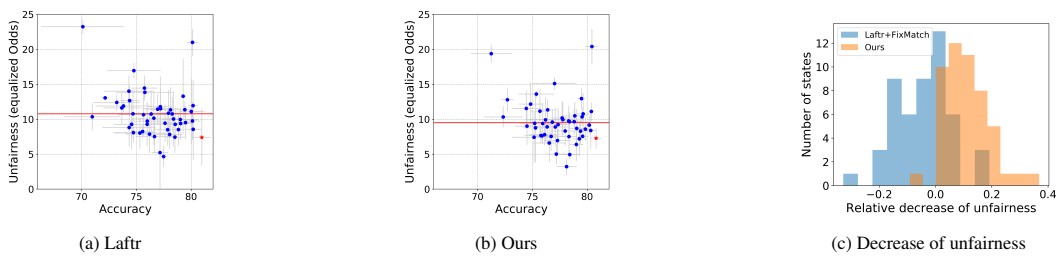

(a) Laftr        (b) Ours        (c) Decrease of unfairness

Figure 5: Unfairness and accuracy tested on NewAdult. CA as the source domain (red star) and other states as the target domain (blue dots). Red lines indicate the average of unfairness. The relative decrease is calculated by comparing with Laftr.

outperforms it remarkably on fairness with a 70% decrease in the variance of group accuracy and a 30% decrease in the equalized odds. We further sweep the weights of losses and draw Pareto frontiers. As shown in Figure 4, our method significantly outperforms others in achieving accuracy and fairness at the same time.

**Evaluation on tabular data.** We further evaluate our method on the NewAdult dataset [19] which contains census data from all states of the United States. We consider gender as the sensitive attribute and do income classification with an MLP as the model. We set CA as the source domain and all the other states as the target domain. We use random perturbation on tabular data (see details in Appendix D) as the transformations. Results are shown in Figure 5. When applied to other states, the fair model trained on CA becomes unfair (Figure 5a). Our method improves the fairness in most states with a slight improvement in accuracy (Figure 5b). Compared with the one without fair consistency regularization, our method achieves better fairness with a decrease in unfairness in most states (Figure 5c).

### 6.3 Ablation Study

**The role of transformation.** We design transformation functions based on our domain knowledge of latent factors. To investigate the importance of transformations, we test a weaker set of transformations, which includes only cropping and flipping, on the UTKFace-FairFace experiment and report the performance in Table 2. Compared with RandAugment in Table 1, consistency under weak transformations leads to a less effective transfer of accuracy since the neighbor generated transformations is much smaller. The limited transformations also restrict the performance of our method on tabular data (see Appendix E). Though the ability to transfer accuracy is limited by weak

| | Source | | | Target | | |
|---|---|---|---|---|---|---|
| | Acc | Unfairness | | Acc | Unfairness | |
| Method | | $V_{acc}$ | $\Delta_{odds}$ | | $V_{acc}$ | $\Delta_{odds}$ |
| Laftr+FixMatch | 94.08±0.70 | 1.64±0.46 | 3.51±1.46 | 77.05±0.26 | 12.23±3.83 | 6.55±1.54 |
| CFair+FixMatch | 94.09±0.33 | 0.97±0.36 | 2.16±0.97 | 77.25±0.21 | 12.93±2.66 | 9.77±0.95 |
| Ours (w/ Laftr) | 94.25±0.22 | 1.06±0.46 | 2.09±0.55 | 77.32±0.21 | 2.35±1.67 | 4.27±1.41 |
| Ours (w/ CFair) | 94.24±0.26 | 1.67±0.38 | 4.43±0.63 | 77.96±0.38 | 3.34±1.08 | 5.70±1.14 |

Table 2: Transfer fairness and accuracy from UTKFace to FairFace with weak transformations

transformations, our method can still make the transfer process fair as there's a significant decrease in unfairness, as shown in Table 2.

**Fair consistency is essential in transferring fairness.** To see whether enhanced consistency improves accuracy and whether unbalanced consistency leads to unfairness as suggested by Theorem 4.1, we evaluate the accuracy and consistency of each group in the UTKFace-FairFace experiment on the target data. The consistency is measured by testing the model's agreement on the outputs under two random transformations. As shown in Figure 6, groups that obtain higher consistency have higher accuracy, which validates the ability of consistency regularization for transferring accuracy. The training methods that use standard consistency regularization (e.g. Laftr+FixMatch) have been observed to be unfair in the target domain. Figure 6 shows that it is because the model has imbalanced consistency across groups. With our fair consistency regularization, the model gains similar consistency for all groups, resulting in similar group accuracy.

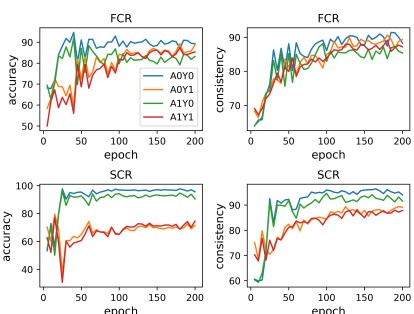

Figure 6: Per-group accuracy and consistency. Compared with the standard consistency regularization (SCR), the model trained with fair consistency regularization (FCR) has more balanced consistency and accuracy.

**The role of components in fair consistency regularization.** Table 3 shows the ablation study. We can see that the consistency in both domains matters. Giving every group the same weight instead of using dynamic weights leads to increased unfairness. Fixing the teacher classifier to be the fair source model, we observe a significant decrease in the accuracy, suggesting the important role of iterative self-training in our algorithm.

| | Acc | Unfairness | |
|---|---|---|---|
| Method | | $V_{acc}$ | $\Delta_{odds}$ |
| Ours | 85.52±0.40 | 2.82±0.87 | 5.70±0.52 |
| w/o consistency in target | 82.43±1.05 | 6.80±1.30 | 5.85±0.40 |
| w/o consistency in source | 82.5±1.58 | 6.63±0.71 | 8.18±1.27 |
| w/o dynamic weights | 84.34±0.19 | 6.86±0.50 | 7.68±0.81 |
| w/o updating $g_{tc}$ | 79.13±0.52 | 3.49±0.63 | 6.65±1.31 |

Table 3: Ablation study on UTKFace-FairFace task

## 7 Conclusion

In this paper, we explore how to transfer fairness under distribution shifts. We derive a sufficient condition and present a theory-guided self-training algorithm based on an intra-group expansion assumption. The key component of our algorithm is fair consistency regularization. We simulate different types of distribution shifts with a synthetic dataset and examine our theoretical findings with it. Abundant experiments with synthetic data and real data have shown that our method has superior performance in transferring fairness and accuracy. Like other self-training methods, one limitation of our method is the reliance on a well-defined data transformation set. Future work will relax this limitation for application to more real-world problems.

## Acknowledgements

This work is supported by National Science Foundation NSF-IIS-FAI program, DOD-ONR-Office of Naval Research, DOD-DARPA-Defense Advanced Research Projects Agency Guaranteeing AI Robustness against Deception (GARD), and Adobe, Capital One and JP Morgan faculty fellowships.

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
