# Supplementary Material

## A  More on Related Work

**Fair machine learning.** Generally, fair machine learning methods fall into three categories: pre-processing, in-processing, and post-processing [44, 7]. In this paper, we focus on in-processing methods that modify learning algorithms to remove discrimination during the training process. As for fair classification, several approaches have been proposed including fair representation learning [66, 41, 4, 67, 42, 56, 17, 73], fairness-constrained optimization [20, 1], causal methods [34, 24, 45], and many other approaches with different techniques [8, 13, 25]. All of those works are for in-distribution fairness, and we investigate out-of-distribution fairness in this paper. We use LAFTR [42], an adversarial learning method that shows advanced performance on fairness [47], to learn a fair model in the source domain and adapt it to the target domain. We also test CFair[72] in our experiments. Many metrics of fairness have been proposed [15] including demographic parity [6], equalized opportunity, and equalized odds [27] which are most widely adopted. In this paper, we use equalized odds to measure unfairness in both domains.

**Distribution shifts.** In many real-world applications, distribution shifts are unavoidable. The goal of existing work addressing distribution shifts is simply to transfer accuracy. [32] propose a benchmark of in-the-wild datasets to study the real distribution shifts. We follow their category of distribution shifts, including subpopulation shifts and domain shifts. Their empirical results on many state-of-the-art methods show that self-training outperforms others on image datasets significantly while having limited performance due to the limited data augmentation on non-image modalities [49]. This finding aligns with our experimental results. [62] conduct a fine-grained analysis of various distribution shifts based on an underlying data generation assumption similar to ours. They also use 3dshapes dataset to simulate different types of distribution shifts. Additional to accuracy, we aim to transfer fairness at the same time in this paper.

**Domain adaptation and self-training.** Inspired by the theoretical work [2], numerous distribution matching approaches have emerged for domain adaption over the past decade. Domain-adversarial training [22] and many of its variants [58, 40, 28, 57] that aim at matching the distribution of two domains in the feature space have shown encouraging results in many applications. However, recent studies [63, 74, 36] show that such methods may fail in many cases since they only optimize part of the theoretical bound. We test DANN [22], and MMD [39], two distribution matching methods in our experiments, and also find them less effective in transferring accuracy and fairness. Recently, another line of work that uses self-training draws increasing attention [70, 3]. Those methods enjoy guarantees [61, 5] and demonstrate superior empirical results with desirable properties such as robustness to spurious features [33, 11, 38] and robustness to dataset imbalance [37]. However, all of those work on domain adaptation only aims at transferring accuracy. Although there is work that studies fairness issues in current domain adaptation methods [35] and proposes to alleviate it by balancing the data [29, 60, 76], fair domain adaptation is still under-explored. Based on the findings that the model's consistency to input transformations is important to generalization [75] and is a core component of self-training [52, 55, 26], we improve the consistency regularization in [55] to achieve fair transferring.

## B  Proof and More Discussion of Fairness under Distribution Shifts

**Lemma B.1.** *Under Assumption 1, for a subpopulation shift that is caused by the shift of the marginal distribution of factor $Y^i$, we have $\mathbb{P}_S(X|Y^i = y^i) = \mathbb{P}_T(X|Y^i = y^i), \forall y^i \in \mathcal{Y}^i$.*

*Proof.* Under Assumption 1, $\mathbb{P}_S(X|Y^{1:K} = \boldsymbol{y}^{1:K}) = \mathbb{P}_T(X|Y^{1:K} = \boldsymbol{y}^{1:K})$. Since the shift only happens on the factor $Y^i$, the marginal distribution of other factors remains the same in the two domains, $\mathbb{P}_S(Y^{\{1:K\}\setminus i}) = \mathbb{P}_T(Y^{\{1:K\}\setminus i})$ where we use $\{1:K\}\setminus i$ to denote $1, .., i-1, i+1, ..., K$.

Then

$$\mathbb{P}_S(X|Y^i = y^i) = \sum_{\boldsymbol{y}^{\{1:K\}\backslash i}} \mathbb{P}_S(X, Y^{\{1:K\}\backslash i} = \boldsymbol{y}^{\{1:K\}\backslash i}|Y^i = y^i)$$

$$= \sum_{\boldsymbol{y}^{\{1:K\}\backslash i}} \mathbb{P}_S(Y^{\{1:K\}\backslash i} = \boldsymbol{y}^{\{1:K\}\backslash i})\mathbb{P}_S(X|Y^i = y^i, Y^{\{1:K\}\backslash i} = \boldsymbol{y}^{\{1:K\}\backslash i})$$

$$= \sum_{\boldsymbol{y}^{\{1:K\}\backslash i}} \mathbb{P}_T(Y^{\{1:K\}\backslash i} = \boldsymbol{y}^{\{1:K\}\backslash i})\mathbb{P}_T(X|Y^i = y^i, Y^{\{1:K\}\backslash i} = \boldsymbol{y}^{\{1:K\}\backslash i})$$

$$= \mathbb{P}_T(X|Y^i = y^i)$$

where the second line holds because of the independence of the latent factors $Y^1, ..., Y^K$. □

Now, we restate Proposition 3.1 and provide the proof.

**Proposition B.1** (Transfer of fairness under subpopulation shift). *Consider the subpopulation shift that is caused by the shifted marginal distribution of a nuisance factor $Y^i$ (i.e., $\mathbb{P}_S(Y^i) \neq \mathbb{P}_T(Y^i)$), while $\mathcal{Y}_S^i = \mathcal{Y}_T^i = \mathcal{Y}^i$. If model $f$ is strictly fair in the source domain under any value of factor $Y^i$ satisfying $\mathbb{P}_S(f(X) = y^l|Y^a = 0, Y^l = y^l, Y^i = y^i) = \mathbb{P}_S(f(X) = y^l|Y^a = 1, Y^l = y^l, Y^i = y^i), \forall y^i \in \mathcal{Y}^i, y^l \in \{0, 1\}$, then $f$ is also fair in target domain with $\Delta_{odds} = 0$.*

*Proof.* In the target domain, the equalized odds (unfairness) is defined as

$$\Delta_{odds} = \frac{1}{2} \sum_{y^l=0}^{1} \left| \mathbb{P}_T(f(X) = y^l|Y^a = 0, Y^l = y^l) - \mathbb{P}_T(f(X) = y^l|Y^a = 1, Y^l = y^l) \right|.$$

Since all latent factors are independent, we have

$$\mathbb{P}_T\left(f(X) = y^l|Y^a = 0, Y^l = y^l\right) = \sum_{y^i \in \mathcal{Y}^i} \mathbb{P}_T(Y^i = y^i)\mathbb{P}_T\left(f(X) = y^l|Y^a = 0, Y^l = y^l, Y^i = y^i\right)$$

and

$$\mathbb{P}_T\left(f(X) = y^l|Y^a = 1, Y^l = y^l\right) = \sum_{y^i \in \mathcal{Y}^i} \mathbb{P}_T(Y^i = y^i)\mathbb{P}_T\left(f(X) = y^l|Y^a = 1, Y^l = y^l, Y^i = y^i\right).$$

Therefore, the $\Delta_{odds}$ in the target domain can be decomposed into

$$\Delta_{odds} = \frac{1}{2} \sum_{y^l=0}^{1} \Big| \sum_{y^i \in \mathcal{Y}^i} \mathbb{P}_T(Y^i = y^i)\big(\mathbb{P}_T\left(f(X) = y^l|Y^a = 0, Y^l = y^l, Y^i = y^i\right)$$

$$- \mathbb{P}_T\left(f(X) = y^l|Y^a = 1, Y^l = y^l, Y^i = y^i\right)\big)\Big|.$$

Since two domains share the same underlying data generative model, and the distribution shift is caused by the shift of the marginal distribution of factor $Y^i$, from Lemma B.1, we have

$$\mathbb{P}_T\left(X|Y^a = 0, Y^l = y^l, Y^i = y^i\right) = \mathbb{P}_S\left(X|Y^a = 0, Y^l = y^l, Y^i = y^i\right)$$

and

$$\mathbb{P}_T\left(X|Y^a = 1, Y^l = y^l, Y^i = y^i\right) = \mathbb{P}_S\left(X|Y^a = 1, Y^l = y^l, Y^i = y^i\right).$$

Thus the conditional distribution of the model's prediction also remains, as

$$\mathbb{P}_T\left(f(X)|Y^a = 0, Y^l = y^l, Y^i = y^i\right) = \mathbb{P}_S\left(f(X)|Y^a = 0, Y^l = y^l, Y^i = y^i\right)$$

and

$$\mathbb{P}_T\left(f(X)|Y^a = 1, Y^l = y^l, Y^i = y^i\right) = \mathbb{P}_S\left(f(X)|Y^a = 1, Y^l = y^l, Y^i = y^i\right).$$

In this case, if the source model is strictly fair that $\forall y^i \in \mathcal{Y}^i, y^l \in \{0, 1\}$ the following holds

$$\mathbb{P}_S(f(X) = y^l|Y^a = 0, Y^l = y^l, Y^i = y^i) = \mathbb{P}_S(f(X) = y^l|Y^a = 1, Y^l = y^l, Y^i = y^i),$$

then it is also fair in the target domain with

$$\Delta_{odds} = \frac{1}{2} \sum_{y^l=0}^{1} \Big| \sum_{y^i \in \mathcal{Y}^i} \mathbb{P}_T(Y^i = y^i)\big(\mathbb{P}_S\left(f(X) = y^l|Y^a = 0, Y^l = y^l, Y^i = y^i\right)$$

$$- \mathbb{P}_S\left(f(X) = y^l|Y^a = 1, Y^l = y^l, Y^i = y^i\right)\big)\Big| = 0.$$

□

This proposition explains why the fair source model is also fair in the target domain under Sshift 1 in our experiments (see Section 6.1). In addition to shifts of nuisance factors, the subpopulation shifts can also be caused by the marginal distribution shift of the label and sensitive attribute. The following proposition argues that the fair model is also in the target domain under such distribution shifts.

**Proposition B.2** (Transfer of fairness under subpopulation shift of sensitive attribute). *Consider the subpopulation shift that is caused by the shifted marginal distribution of sensitive attribute $Y^a$ (i.e., $\mathbb{P}_S(Y^a) \neq \mathbb{P}_T(Y^a)$), while $\mathcal{Y}_S^a = \mathcal{Y}_T^a = \mathcal{Y}^a = \{0, 1\}$. If model $f$ is fair in the source domain with $\Delta_{odds}^S = 0$, then it is also fair in the target domain with $\Delta_{odds}^T = 0$.*

*Proof.* The proof is similar to the previous one. Since

$$\Delta_{odds}^S = \frac{1}{2} \sum_{y^l=0}^{1} \left| \mathbb{P}_S(f(X) = y^l | Y^a = 0, Y^l = y^l) - \mathbb{P}_S(f(X) = y^l | Y^a = 1, Y^l = y^l) \right|,$$

and from Lemma B.1 we know that

$$\mathbb{P}_S(X | Y^a = 0, Y^l = y^l) = \mathbb{P}_T(X | Y^a = 0, Y^l = y^l)$$
$$\mathbb{P}_S(X | Y^a = 1, Y^l = y^l) = \mathbb{P}_T(X | Y^a = 1, Y^l = y^l),$$

thus,

$$\Delta_{odds}^T = \frac{1}{2} \sum_{y^l=0}^{1} \left| \mathbb{P}_T(f(X) = y^l | Y^a = 0, Y^l = y^l) - \mathbb{P}_T(f(X) = y^l | Y^a = 1, Y^l = y^l) \right|$$

$$= \frac{1}{2} \sum_{y^l=0}^{1} \left| \mathbb{P}_S(f(X) = y^l | Y^a = 0, Y^l = y^l) - \mathbb{P}_S(f(X) = y^l | Y^a = 1, Y^l = y^l) \right|$$

$$= \Delta_{odds}^S = 0.$$

$\square$

Such a result also holds for subpopulation shifts caused by the shift of label $Y^l$. This proposition explains why the fair source model is also fair in the target domain under Sshift 2 in our experiments (see Section 6.1). It suggests that encouraging fairness is able to alleviate spurious correlation. We leave more studies on this interesting finding to future work.

**Remark.** All the above analyses are based on the population distribution where $\mathbb{P}_S(f(X) = y^l | Y^a = 0, Y^l = y^l) = \mathbb{E}_{\mathbb{P}_S(X, Y^{1:K})}(f(X) = y^l | Y^a = 0, Y^l = y^l)$. In practice, it is estimated by finite samples. Insufficient samples would cause estimation errors in fairness and bring another challenge for transferring fairness. In this paper, we only consider the fairness measured by population distribution. Future work will investigate the impact of estimation error on transferring fairness and the way to resolve it.

## C   Proof of the Sufficient Condition for Transferring Fairness

Our proof is based on the theory in [61].

**Theorem C.1.** *(Restatement of Lemma A.8 in [61]) We assume that $U_a^y$ satisfies $(\bar{\alpha}, \bar{c})$-multiplicative expansion for $\varepsilon_{U_a^y}(g_{tc}) \leq \bar{\alpha} < 1/3$ and $\bar{c} > 3$. We define $c \triangleq \min\{1/\bar{\alpha}, \bar{c}\}$. Then for any classifier $g : \mathcal{X} \to \mathcal{Y}$, the error of it on the group $U_a^y$ is upper bounded as:*

$$\varepsilon_{U_a^y}(g) \leq \frac{c+1}{c-1} L_{U_a^y}(g, g_{tc}) + \frac{2c}{c-1} R_{U_a^y}(g) - \varepsilon_{U_a^y}(g_{tc})$$

**Theorem C.2.** *(A restricted version of the above theorem) We assume that $U_a^y$ satisfies $(\bar{\alpha}, \bar{c})$-multiplicative expansion for $\varepsilon_{U_a^y}(g_{tc}) \leq \bar{\alpha} < 1/3$ and $\bar{c} > 3$. We define $c \triangleq \min\{1/\bar{\alpha}, \bar{c}\}$. Then for any classifier $g : \mathcal{X} \to \mathcal{Y}$ satisfies $L_{U_a^y}(g, g_{tc}) \leq \varepsilon_{U_a^y}(g_{tc})$, the error of it on the group $U_a^y$ is upper bounded as:*

$$\varepsilon_{U_a^y}(g) \leq \frac{2}{c-1} \varepsilon_{U_a^y}(g_{tc}) + \frac{2c}{c-1} R_{U_a^y}(g)$$

*Proof.*

$$\varepsilon_{U_a^y}(g) \leq \frac{c+1}{c-1} L_{U_a^y}(g, g_{tc}) + \frac{2c}{c-1} R_{U_a^y}(g) - \varepsilon_{U_a^y}(g_{tc})$$

$$\varepsilon_{U_a^y}(g) \leq \frac{2}{c-1} \varepsilon_{U_a^y}(g_{tc}) + \frac{2c}{c-1} R_{U_a^y}(g) \qquad \text{(because } L_{U_a^y}(g, g_{tc}) \leq \varepsilon_{U_a^y}(g_{tc}))$$

$\square$

**Theorem C.3.** *If* $L_{U_a^y}(g, g_{tc}) \leq \varepsilon_{U_a^y}(g_{tc})$*, we have*
$$\varepsilon_{U_a^y}(g) \geq \varepsilon_{U_a^y}(g_{tc}) - L_{U_a^y}(g, g_{tc})$$

*Proof.* By triangle inequality. $\square$

Now, we restate Theorem 4.1 and provide the proof.

**Theorem C.4.** *Suppose we have a teacher classifier $g_{tc}$ with bounded unfairness such that $|\varepsilon_{U_a^y}(g_{tc}) - \varepsilon_{U_{a'}^{y'}}(g_{tc})| \leq \gamma, \forall a, a' \in \mathcal{A}$ and $y, y' \in \mathcal{Y}$. We assume intra-group expansion where $U_a^y$ satisfies $(\bar{\alpha}, \bar{c})$-multiplicative expansion and $\varepsilon_{U_a^y}(g_{tc}) \leq \bar{\alpha} < 1/3$ and $\bar{c} > 3, \forall a, y$. We define $c \triangleq \min\{1/\bar{\alpha}, \bar{c}\}$, and set $\mu \leq \varepsilon_{U_a^y}(g_{tc}), \forall a, y$. If we train our classifier with the algorithm*

$$\min_{g \in G} \max_{a,y} R_{U_a^y}(g) \tag{7}$$
$$\text{s.t.} \quad L_{U_a^y}(g, g_{tc}) \leq \mu \quad \forall a, y$$

*then the error and unfairness of the optimal solution $\hat{g}$ on the distribution $U$ are bounded with*

$$\varepsilon(\hat{g}) \leq \frac{2}{c-1} \varepsilon_U(g_{tc}) + \frac{2c}{c-1} R_U(\hat{g}),$$

$$\Delta_{odds}(\hat{g}) \leq \frac{2}{c-1} (\gamma + \mu + c \max_{a,y} R_{U_a^y}(\hat{g})).$$

*Proof.* The upper bound of error is derived from Theorem C.2. For the unfairness, by definition
$$\Delta_{odds}(\hat{g}) = \frac{1}{2} \left( \left| \varepsilon_{U_0^0}(\hat{g}) - \varepsilon_{U_1^0}(\hat{g}) \right| + \left| \varepsilon_{U_0^1}(\hat{g}) - \varepsilon_{U_1^1}(\hat{g}) \right| \right).$$
Based on the upper bound of group error from Theorem C.2, and the lower bound of it from Theorem C.3, we have

$$\left| \varepsilon_{U_0^0}(\hat{g}) - \varepsilon_{U_1^0}(\hat{g}) \right|$$
$$\leq \max \left\{ \frac{2}{c-1}\gamma + \frac{2}{c-1} L_{U_1^0}(\hat{g}, g_{tc}) + \frac{2c}{c-1} R_{U_0^0}(\hat{g}), \right.$$
$$\left. \frac{2}{c-1}\gamma + \frac{2}{c-1} L_{U_0^0}(\hat{g}, g_{tc}) + \frac{2c}{c-1} R_{U_1^0}(\hat{g}) \right\} \qquad \text{(because } c > 3)$$
$$= \frac{2}{c-1}\gamma + \frac{2}{c-1} \max \left\{ L_{U_1^0}(\hat{g}, g_{tc}) + c R_{U_0^0}(\hat{g}), L_{U_0^0}(\hat{g}, g_{tc}) + c R_{U_1^0}(\hat{g}) \right\}$$
$$\leq \frac{2}{c-1}(\gamma + \mu + c \max_a R_{U_a^0}(\hat{g})).$$

Therefore,

$$\Delta_{odds}(\hat{g}) \leq \frac{2}{c-1} \left( \gamma + \mu + \frac{c}{2} \left( \max_a R_{U_a^0}(\hat{g}) + \max_a R_{U_a^1}(\hat{g}) \right) \right)$$
$$\leq \frac{2}{c-1}(\gamma + \mu + c \max_{a,y} R_{U_a^y}(\hat{g})).$$

$\square$

**Upper bound of $V_{acc}$.** From Theorem C.2 we know that the group accuracy is upper bounded by $\varepsilon_{U_a^y}(\hat{g}) \leq \frac{2}{c-1} \varepsilon_{U_a^y}(g_{tc}) + \frac{2c}{c-1} R_{U_a^y}(\hat{g})$. The variance of group accuracy is defined as
$$V_{acc}(\hat{g}) = Var(\{\mathbb{P}(\hat{Y} = y | A = a, Y = y), \forall a, y\})$$
$$= Var(\{\varepsilon_{U_a^y}(\hat{g}), \forall a, y\})$$

If we assume the same estimation error for all the groups when we use the upper bound to estimate the group accuracy with $\varepsilon_{U_a^y}(\hat{g}) = \frac{2}{c-1}\varepsilon_{U_a^y}(g_{tc}) + \frac{2c}{c-1}R_{U_a^y}(\hat{g})$, then

$$V_{acc}(\hat{g}) = Var\left(\left\{\frac{2}{c-1}\varepsilon_{U_a^y}(g_{tc}) + \frac{2c}{c-1}R_{U_a^y}(\hat{g}), \forall a, y\right\}\right)$$

When the teacher classifier has bounded unfairness with $|\varepsilon_{U_a^y}(g_{tc}) - \varepsilon_{U_{a'}^{y'}}(g_{tc})| \leq \gamma, \forall a, a', y, y'$, the variance of group accuracy would be mainly affected by the variance of group consistency loss $Var(\{R_{U_a^y}(\hat{g}), \forall a, y\})$. Therefore, this upper bound also suggests us to balance the consistency loss while minimizing it.

**Multi-sensitive attribute and multi-class cases.** It is obvious that the Theorem 4.1 still holds for the binary-sensitive attribute and multi-class case where $\mathcal{Y} = \{1, 2, .., M\}$. For the multi-sensitive attribute case, the key problem is how to define the unfairness. If we define the equalized odds in general cases to be the following one, then it is easy to see that Theorem 4.1 still holds.

$$\Delta_{odds}(\hat{g}) = \frac{1}{|\mathcal{Y}|}\sum_{y \in \mathcal{Y}}\max_{a, a' \in \mathcal{A}}\left|\varepsilon_{U_a^y}(\hat{g}) - \varepsilon_{U_{a'}^y}(\hat{g})\right|$$

# D  Details of Experiments

## D.1  Synthetic Dataset

The 3dshapes dataset [2] [31] contains 480000 RGB images (the size is $64 \times 64 \times 3$) of 3D objects. Every image is generated by six latent factors (*shape*, *object hue*, *scale*, *orientation*, *floor hue*, *wall hue*) which are annotated along with images. The sample spaces of these factors are shown in Table 4.

| Factor | Sample space |
|---|---|
| shape | 4 values in [0, 1, 2, 3] |
| object hue | 10 values linearly spaced in [0, 1] |
| scale | 8 values linearly spaced in [0, 1] |
| orientation | 15 values linearly spaced in [-30, 30] |
| floor hue | 10 values linearly spaced in [0, 1] |
| wall hue | 10 values linearly spaced in [0, 1] |

Table 4: Latent factors in 3dshapes dataset.

**How to simulate different types of distribution shift?** By varying the marginal distribution of latent factors and then sample images according to the distribution of latent factors, we can simulate different distribution shifts. In this paper, we set the image as input $X$, and set class $Y = shape$, sensitive attribute $A = object\ hue$, and a nuisance factor that might shift to be $D = scale$. We consider a binary case, and restrict the *shape* to be in $\{0, 1\}$ (i.e. $\{$cube, cylinder$\}$) and *object hue* to be in $\{0, 1\}$ (i.e. $\{$red, yellow$\}$). In our experiments, we simulate four types of distribution shift. Their specific settings are shown in Table 5. We show examples from two domains under different shifts in Figure 7.

## D.2  Real Datasets

**UTKFace**[3] [71] is a face dataset with images annotated with age, gender, and race. The data is collected from MORPH, CACD and Web. In our experiments, we use the aligned and cropped face images with ages larger than 10. We do gender classification which is a binary classification task, and set sensitive attribute to be the race. We consider binary-sensitive attribute case in our experiment by setting race to be white or non-white. The statistics of this dataset are shown in Table 6.

---

[2]https://github.com/deepmind/3d-shapes
[3]https://susanqq.github.io/UTKFace/

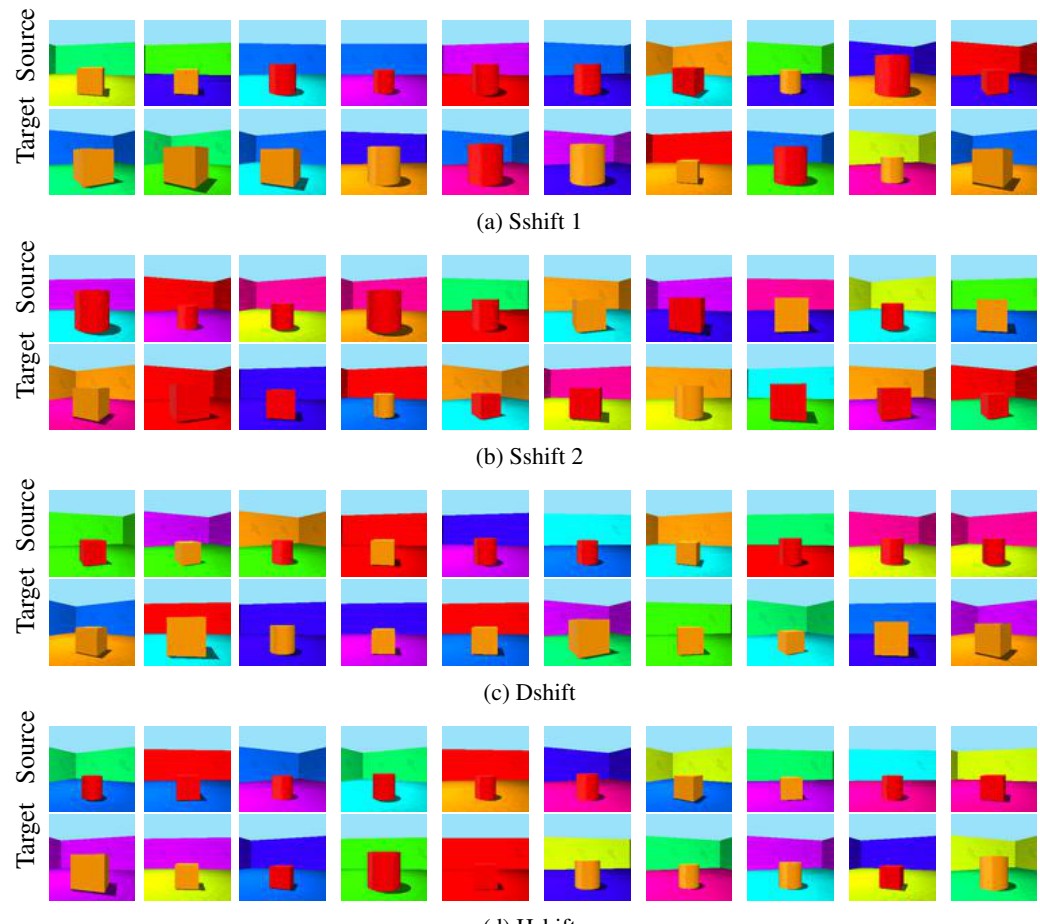

Figure 7: Randomly sampled examples from two domains under different distribution shifts.

| | Factor | Source | Target |
|---|---|---|---|
| Sshift 1 | $\mathbb{P}(Y, A)$ | $[0.1, 0.4, 0.4, 0.1]$ | same |
| | $\mathbb{P}(D)$ | $[\frac{4}{16}, \frac{4}{16}, \frac{3}{16}, \frac{1}{16}, \frac{1}{16}, \frac{1}{16}, \frac{1}{16}, \frac{1}{16}]$ | $[\frac{1}{16}, \frac{1}{16}, \frac{1}{16}, \frac{1}{16}, \frac{1}{16}, \frac{3}{16}, \frac{4}{16}, \frac{4}{16}]$ |
| Sshift 2 | $\mathbb{P}(Y, A)$ | $[0.1, 0.4, 0.4, 0.1]$ | $[0.4, 0.1, 0.1, 0.4]$ |
| | $\mathbb{P}(D)$ | $[\frac{1}{8}, \frac{1}{8}, \frac{1}{8}, \frac{1}{8}, \frac{1}{8}, \frac{1}{8}, \frac{1}{8}, \frac{1}{8}]$ | same |
| Dshift | $\mathbb{P}(Y, A)$ | $[0.1, 0.4, 0.4, 0.1]$ | same |
| | $\mathbb{P}(D)$ | $[\frac{1}{2}, \frac{1}{2}, 0, 0, 0, 0, 0, 0]$ | $[\frac{1}{8}, \frac{1}{8}, \frac{1}{8}, \frac{1}{8}, \frac{1}{8}, \frac{1}{8}, \frac{1}{8}, \frac{1}{8}]$ |
| Hshift | $\mathbb{P}(Y, A)$ | $[0.1, 0.4, 0.4, 0.1]$ | $[0.4, 0.1, 0.1, 0.4]$ |
| | $\mathbb{P}(D)$ | $[\frac{1}{2}, \frac{1}{2}, 0, 0, 0, 0, 0, 0]$ | $[\frac{1}{8}, \frac{1}{8}, \frac{1}{8}, \frac{1}{8}, \frac{1}{8}, \frac{1}{8}, \frac{1}{8}, \frac{1}{8}]$ |

Table 5: Simulate different distribution shifts. $\mathbb{P}(Y, A)$ is represented by the proportions of four groups as $[\mathbb{P}(Y = 0, A = 0), \mathbb{P}(Y = 0, A = 1), \mathbb{P}(Y = 1, A = 0), \mathbb{P}(Y = 1, A = 1)]$. $\mathbb{P}(D)$ is represented by the proportions of eight possible values of *scale*. Other factors have uniform marginal distributions. Images in two domains are sampled according to the marginal distributions of six latent factors.

**FairFace**[4] [30] is another large-scale face dataset with images annotated with age, gender, and race as well. Different from UTKFace, the data in FairFace is collected from Flickr, Twitter, Newspapers,

---

[4]https://github.com/joojs/fairface

| $(Y, A)$ | | (Male, White) | (Male, Black) | (Female, White) | (Female, Black) |
|---|---|---|---|---|---|
| UTK (S) | train | 3127 | 1508 | 2480 | 1450 |
| | test | 1377 | 651 | 1027 | 617 |
| FairFace (T) | train | 7796 | 4650 | 6946 | 5160 |
| | test | 984 | 620 | 839 | 635 |

Table 6: Statistics of UTKFace and FairFace datasets.

and the Web. We also use images with ages larger than 10 in our experiments. We set the label to be the gender and the sensitive attribute to be the race. See statistics in Table 6. All face images in UTK and FairFace are resized to $96 \times 96 \times 3$ in our experiments.

**NewAdult**[5] [19] is a suite of datasets derived from US Census surveys. The data spans multiple years and all states of the United States which is a good fit for studying distribution shifts. In our experiments, we use 2018 data that span all states and do income classification with a threshold of 50,000 dollars. We set gender to be the sensitive attribute. We consider a problem that we train a fair classifier in California (source domain) and deploy it in other states (target domain). The statistics are shown in Table 7. The input contains 10 features (see Appendix B.1 in [19]) which are preprocessed to one-hot embeddings in our experiments.

| $(Y, A)$ | | (High, Male) | (High, Female) | (Low, Male) | (Low, Female) |
|---|---|---|---|---|---|
| CA (S) | train | 33258 | 22314 | 39224 | 42169 |
| | test | 14839 | 9924 | 15990 | 17947 |
| Other states (T) | train | 232162 | 140876 | 296826 | 351970 |
| | test | 101934 | 57798 | 127654 | 150544 |

Table 7: Statistics of NewAdult dataset.

## D.3 Experimental Settings

### D.3.1 Experiments on 3dshapes

**Model.** We use a two-layer MLP with 512 hidden units as the encoder and one linear layer as the classifier. The adversaries used in Laftr, CFair, and DANN are also two-layer MLP with 512 hidden units. ReLU is used as the activation function.

**Transformations.** We use random center cropping and padding as the transformation functions in consistency regularization. Such transformations can perfectly change the *scale* of the objects to propagate labels from the source domain to the target domain.

**Setup.** We use SGD as the optimizer. We train every model with 200 epochs and select the best model according to the model's performance on the validation set. Base and Laftr only have access to the source data, and the model selection is based on the source validation set. For other methods that can access the unlabeled target data, the model selection is based on the labeled target validation set. Since *accuracy* and *fairness* are both important metrics, we select the best model according to the value of accuracy minus unfairness which is $Acc - \Delta_{odds}$. The coefficients of the fairness loss and consistency loss are both set to be 1. We run every method five times and report the mean and the standard deviation.

---

[5]https://github.com/zykls/folktables

### D.3.2 Experiments on UTK-FairFace

**Model.** We use VGG16 and ResNet18 as the model in our experiments. The last linear layer is the classifier, and all the previous layers construct the encoder. When we use VGG16 as the model, we set every adversary used in Laftr, CFair, and DANN to be a two-layer MLP with 1024 hidden neurons. When ResNet18 is the model, the adversary has 512 hidden neurons.

**Transformations.** We use RandAugment [18] as the transformation function which contains data augmentations that are the best for the CIFAR-10 dataset. To restrict the transformations to be group-preserving, we exclude augmentations that may change the color (so to change the race). The transformation function used in our experiments contains AutoContrast, Brightness, Equalize, Identity, Posterize, Rotate, Sharpness, ShearX, ShearY, TranslateX, and TranslateY. In this experiment, we use a weak augmented (with random cropping and flipping) image as the original input $x$ and a strong augmented (with RandAugment) image as the transformed input $t(x)$.

**Setup.** We use SGD as the optimizer. We train every model with 200 epochs and use the way introduced in the 3dshapes experiment to select the best model. The coefficients of the fairness loss and consistency loss are both set to be 1. We run every method five times and report the mean and the standard deviation.

### D.3.3 Experiments on NewAdult

**Model.** We use a 3-layer MLP with hidden sizes of (256, 512, 256) as the encoder and a 2-layer MLP with a hidden size of 128 as the classifier. Every adversary is a two-layer MLP with 128 hidden neurons.

**Transformations.** Studies on data augmentations for tabular data are very limited. In this paper, we use random corruptions on the input features as the transformation function. There are ten features in the input, and every time we only corrupt half of them. Additionally, for important factors that are highly correlated with the label, including the OCCP (occupation), COW (class of worker), we do not do any corruption. For factor SEX (gender), we do not do any corruption to preserve the group. For continuous factors including AGEP (age), SCHL (educational attainment), and WKHP (work hours), we do perturbations within a range. For other factors, we do uniformly sampling from their value spaces as corruptions. We do such transformations based on our assumption that they do not change the label. For example, two individuals that have five years of age gap but have the same other features should have similar income, and two individuals that only differ in the place of birth should have similar income. We admit that such transformations may not be the best ones. We need better domain knowledge on income prediction to design more powerful transformations. We leave the improvement of transformations for tabular data to future work.

**Setup.** We use SGD as the optimizer. We train every model with 200 epochs and use the metric introduced in the 3dshapes experiment to select the best model. The coefficients of the fairness loss and consistency loss are both set to be 1. We run every method five times and report the mean and the standard deviation.

## D.4 Baselines

**Laftr** is an adversarial learning method for algorithmic fairness. The adversary aims at accurately predicting the sensitive attribute based on the representation, while the encoder aims at making it hard. By adversarial learning, the representation will not contain information on sensitive attributes, so the prediction based on it will be fair. We denote the data in each group to be $\mathcal{D}_a^y = \{x \in \mathcal{D} | A = a, Y = y\}$. The fairness loss is designed to be

$$L_{fair} = \sum_{(a,y) \in \{0,1\}^2} \frac{1}{|\mathcal{D}_a^y|} \sum_{x \in \mathcal{D}_a^y} |h(f(x)) - a|.$$

where $f$ is the encoder, $h$ is the adversary. [42] prove that this loss is an upper bound of the equalized odds. The adversary minimizes this loss, while the encoder maximizes this loss with a gradient reversal layer.

**CFair** is similar to Laftr but uses two adversaries $h'$, and $h''$ for two classes with a balanced error rate (BER) defined as follows. We denote the data from one class to be $\mathcal{D}^y = \{x \in \mathcal{D} | Y = y\}$.

$$L_{fair} = \text{BER}_{\mathcal{D}^0}(h'(f(x)) \parallel A) + \text{BER}_{\mathcal{D}^1}(h''(f(x)) \parallel A)$$

where $\text{BER}_{\mathcal{D}^0}(h'(f(\boldsymbol{x})) \parallel A) = \frac{1}{2}\mathbb{P}_{\mathcal{D}^0}(h'(f(\boldsymbol{x})) \neq A|A = 0) + \frac{1}{2}\mathbb{P}_{\mathcal{D}^0}(h'(f(\boldsymbol{x})) \neq A|A = 1)$. In practice, the balanced error rate is estimated by the following cost-sensitive cross-entropy loss.

$$\mathbb{P}_{\mathcal{D}^0}(h'(f(\boldsymbol{x})) \neq A|A = 0) \leq \frac{\text{CE}_{\mathcal{D}_0^0}(h'(f(\boldsymbol{x})) \parallel A)}{\mathbb{P}_{\mathcal{D}^0}(A = 0)}$$

**Laftr+FixMatch** uses the same framework as our method but with a standard consistency regularization that does not care about group performance. The consistency loss is defined as

$$L_{consis}(g) = \frac{1}{|\mathcal{D}|} \sum_{\boldsymbol{x} \in \mathcal{D}} \mathbb{1}(\max(g_{tc}(\boldsymbol{x})) \geq \tau) H(\arg\max(g_{tc}(\boldsymbol{x})), g(t(\boldsymbol{x})))$$

where $\mathcal{D}$ denotes the entire dataset.

### D.5  Time and Space Complexity

Compared with Base, Laftr, and CFair which only uses labeled source data, our method needs more training time and memory since we use unlabeled target data as well. Compared with other baselines that also use target data, such as Laftr+DANN, the time complexity of our method is comparable to theirs. Nevertheless, our method needs much fewer parameters than Laftr+DANN since it requires an adversary to do domain classification while we do not need it. Our method has the same model parameters as that in Laftr but with an additional consistency loss.

## E  More Experimental Results

### E.1  Additional Results on UTKFace-FairFace with a Different Data Setting

To evaluate our method in extreme circumstances, we conduct the UTKFace-FairFace experiment with less labeled source data and more unlabeled target data (see Table 8). We also consider the race "white" and "non-white". Are shown in Table 9, we get consistent results that our method outperforms all baselines and can effectively transfer accuracy as well as fairness.

| $(Y, A)$ | | (Male, White) | (Male, Non-white) | (Female, White) | (Female, Non-white) |
|---|---|---|---|---|---|
| UTK (S) | train | 1373 | 750 | 1650 | 1227 |
| | test | 565 | 285 | 614 | 370 |
| FairFace (T) | train | 11429 | 16574 | 8024 | 16838 |
| | test | 1712 | 2453 | 1176 | 2518 |

Table 8: Statistics of UTK and FairFace datasets used in Table 9.

| | Source | | | Target | | |
|---|---|---|---|---|---|---|
| | Acc | Unfairness | | Acc | Unfairness | |
| Method | | $V_{acc}$ | $\Delta_{odds}$ | | $V_{acc}$ | $\Delta_{odds}$ |
| Base | 89.93±0.43 | 2.79±0.74 | 4.65±0.44 | 73.48±0.56 | 7.49±3.50 | 6.09±1.07 |
| Laftr | 90.61±0.33 | 1.28±0.43 | 3.62±1.17 | 73.29±0.70 | 5.42±1.33 | 7.78±1.77 |
| CFair | 90.68±0.35 | 1.20±0.59 | 3.61±0.93 | 73.82±0.81 | 5.71±1.54 | 7.37±1.40 |
| Laftr+DANN | 90.53±0.98 | 1.59±0.97 | 4.62±1.24 | 74.44±1.38 | 6.94±1.53 | 10.26±1.85 |
| CFair+DANN | 90.23±0.88 | 1.82±0.97 | 4.96±1.15 | 74.53±1.46 | 9.27±2.16 | 9.96±1.49 |
| Laftr+FixMatch | 95.01±0.10 | 1.37±0.44 | 4.65±1.00 | 83.77±0.45 | 11.58±1.16 | 6.56±1.74 |
| CFair+FixMatch | 95.37±0.24 | 1.13±0.21 | 3.58±0.90 | 83.62±0.51 | 11.96±1.05 | 5.29±1.76 |
| Ours (w/ Laftr) | 94.77±0.33 | 1.35±0.70 | 3.28±0.79 | 84.65±1.13 | 2.92±0.72 | 6.99±0.41 |
| Ours (w/ CFair) | 94.92±0.43 | 1.09±0.30 | 3.00±1.09 | 84.71±1.10 | 3.57±0.60 | 7.34±0.91 |

Table 9: Transfer fairness and accuracy from UTKFace to FairFace with less source data.

| | Source | | | Target | | |
|---|---|---|---|---|---|---|
| | Acc | Unfairness | | Acc | Unfairness | |
| Transformation | | $V_{acc}$ | $\Delta_{odds}$ | | $V_{acc}$ | $\Delta_{odds}$ |
| None | 93.24 | 1.19 | 2.44 | 74.35 | 6.92 | 9.79 |
| All | 96.08 | 0.96 | 2.59 | 85.52 | 2.82 | 5.70 |
| AutoContrast | 94.82 | 1.12 | 2.66 | 79.69 | 5.55 | 7.48 |
| Brightness | 95.61 | 0.95 | 1.48 | 82.16 | 4.89 | 6.39 |
| Color | 95.53 | 1.07 | 1.28 | 81.32 | 6.66 | 8.22 |
| Contrast | 94.93 | 1.31 | 2.29 | 79.35 | 6.37 | 8.39 |
| Equalize | 95.15 | 1.47 | 2.33 | 79.17 | 5.88 | 6.91 |
| Identity | 96.21 | 1.03 | 1.31 | 81.58 | 3.44 | 7.29 |
| Posterize | 94.92 | 1.77 | 3.06 | 79.63 | 5.26 | 6.01 |
| Rotate | 96.13 | 0.72 | 1.83 | 84.33 | 3.80 | 6.34 |
| Sharpness | 95.73 | 1.03 | 2.64 | 81.26 | 5.33 | 7.09 |
| ShearX | 95.45 | 1.70 | 0.99 | 82.47 | 3.30 | 3.72 |
| ShearY | 96.25 | 0.54 | 1.75 | 84.26 | 3.96 | 6.07 |
| Solarize | 95.89 | 0.98 | 2.67 | 80.38 | 7.37 | 8.79 |
| TranslateX | 96.11 | 0.89 | 1.79 | 83.49 | 2.31 | 6.13 |
| TranslateY | 95.53 | 0.97 | 2.83 | 83.04 | 7.16 | 6.17 |

Table 10: Results by using different transformations in our method. Average results of three trials.

## E.2 Additional Results on UTKFace-FairFace with Different Transformations

To investigate the effect of different transformations in our method, we evaluate 14 transformations in RandAugment and report the results in Table 9. All the transformations can improve the accuracy in both domains. The effect on fairness varies. We find that *Solarize*, *Color*, and *TranslateX* increase the unfairness in the source domain the most, and *Contrast*, *Color* and *Solarize* have the highest unfairness in the target domain. Note that, it does not mean that these augmentations always lead to unfairness but that they are not suitable for our method. Recall that our theory and algorithm are built upon the intra-group expansion assumption. Transformations like *Contrast*, *Color*, and *Solarize* may change the sensitive attribute "race" and break this assumption. Thus, in our experiments (Table 1) we use all the transformations excluding *Contrast*, *Color*, and *Solarize*.

## E.3 A Byproduct: Alleviate the Disparate Impact of Semi-supervised Learning

[76] find that semi-supervised learning methods may have a disparate impact. The classes that have high accuracy on labeled data are likely to benefit more from semi-supervised learning on unlabeled data. We test this argument on CIFAR-10 with FixMatch as the semi-supervised learning method. We use ResNet18 as the model. We randomly sample 500 images to be labeled data and treat others as unlabeled data. We use the benefit ratio proposed in [76] as the metric for the benefit of semi-supervised learning, defined as

$$BR(\mathcal{D}) = \frac{a_{semi}(\mathcal{D}) - a_{baseline}(\mathcal{D})}{a_{ideal}(\mathcal{D}) - a_{baseline}(\mathcal{D})}. \tag{8}$$

where $\mathcal{D}$ denotes the data from one class. $a_{semi}(\mathcal{D})$ is the model's test accuracy after semi-supervised learning, $a_{baseline}(\mathcal{D})$ is the test accuracy of the base model that is trained on labeled data, and $a_{ideal}(\mathcal{D})$ is the test accuracy of the ideal model where all data are labeled. We evaluate the benefit

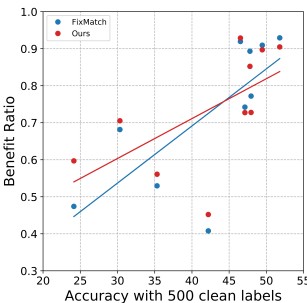

Figure 9: With fair consistency regularization, our method alleviates the disparate impact of FixMatch.

ratio of FixMatch on ten classes. As the blue line in Figure 9 shows, the rich gets richer, and the poor gets poorer after semi-supervised learning. Our method (without using Laftr) can directly apply to this task. By using the proposed fair consistency regularization (red line in Figure 9), we can significantly improve the benefit ratio of the poor classes. Therefore, fair consistency regularization is a byproduct of this paper which is able to alleviate the disparate impact of semi-supervised learning.

## F   Impact and Limitations

The fairness of machine learning is a critical problem in today's real-world applications. When distribution shifts happen, the collapse of fair systems will cause unexpected discrimination, resulting in severe negative social impacts. The fairness that is robust to distribution shifts is essential but is less explored. In this paper, the theoretical analysis of how fairness changes under different distribution shifts sheds light on the deep reasons for the collapse of fairness. The theory-guided self-training algorithm proposed in this paper explores a promising way to tackle distribution shifts. We hope our work will inspire more algorithms for this important and practical task.

The major limitation of our method is that it strongly relies on pre-defined transformations as all the other self-training methods. The transformations are designed to be group-preserving based on our prior knowledge. Our experiments show that self-training with less powerful transformations has limited ability in propagating labels from source to target (i.e. transfer accuracy). Valid transformation functions on image data are thoroughly studied in existing work, while transformations on non-image data such as tabular data are much less explored. Our method with more powerful transformations on tabular data is expected to have significant improvement. Future work is encouraged to relax this limitation, such as by using a generative model as the transformation function.