# OpenReview forum: "Transferring Fairness under Distribution Shifts via Fair Consistency Regularization"
_NeurIPS.cc/2022/Conference — NeurIPS 2022 Accept_

### Official Review · Reviewer_15g2 · 2022-07-03

**Rating:** 4
**Confidence:** 4
**Soundness:** 3 good
**Presentation:** 3 good
**Contribution:** 2 fair

**Summary:**

This paper proposes to use consistency regularization to improve fairness in ML models under distribution shifts. Specifically, the paper extends upon LAFTR (an adversarial learning algorithm for fairness) and FixMatch to impose the consistency constraints via self-training, and for the consistency constraints the authors extend the existing consistency notion with "intra-group expansion", i.e., the transformation/augmentation should be applied within each group and label.

The authors performed experiments on both a synthetic dataset and two real datasets, and show sometimes the fairness in the target domain can be improved compared to existing approaches.

**Questions:**

- How does the result in Table 1 show the superiority of the proposed methods on target fairness? From the numbers it is unclear if the proposed method works better than Laftr+FixMatch or CFair+FixMatch.

- Why do augmentations designed for robustness definitely improve fairness? Would some of the augmentations (other than obvious ones like color jittering) hurt fairness? The authors should perform a more detailed study on this.

**Limitations:**

N/A, this paper aims to improve fairness in ML models.

**Strengths And Weaknesses:**

Strengths:
- Fairness under distribution shift is an important research topic, and this paper proposes to use the consistency regularization as a novel way to tackle this problem.
- In general the paper is clearly written, the method and related work are well presented.

Weaknesses:
- The proposed algorithm seems to only work well on synthetic data (Figure 3). When transferred to real datasets, the experiments results are rather weak.

(1) from UTKFace to FairFace (Table 1), the proposed approach (either with Laftr or CFair) does not achieve the best fairness in terms of equalized odds. For example, Laftr+FixMatch / CFair + FixMathc both achieve good accuracy and lower equalized odds than the proposed approaches. The group accuracy variance does decrease for the proposed approach, but why is that important compared to the fairness metrics? In addition, all the numbers have high standard deviations, are there any significance test done to show the superiority of some of the methods?

(2) on the tabular dataset, the only results shown are in Figure 4. Other than LAFTR, no other baselines are shown, how do we know if the proposed methods work better than other baselines? Even just compared to LAFTR, from the figures (a) and (b) it is very hard to tell if the proposal made the results more fair (as most data points are almost in the same range, and there is no one-to-one correspondence). For Figure 4(c), there is no significance test either, so it is unclear if the results are significant.

- The transformation/augmentation is an important piece in the overall framework, but it is not discussed in detail at all in this paper. The authors only briefly mentioned what transformations are excluded in Section 6.3. As the authors reuse existing augmentations designed for robustness (not necessarily for fairness), they should perform a more detailed study on how each of those transformations affect fairness, and if the effect is positive or negative.

Minor:
- Figure 3 (a), LAFTER -> LAFTR

---

> ### Author Response · Authors · 2022-08-02
> **$\newcommand{Rc}{\textcolor{blue}{15g2}}$ To Reviewer $\Rc$ (1/2)**
>
> $\newcommand{Rc}{\textcolor{blue}{15g2}}$
>
> Thank you for reviewing our paper and providing valuable feedback. We are glad that you find our paper novel and well-written. We address your questions below in detail.
>
> ---
> > **W1**: The proposed algorithm seems to only work well on synthetic data (Figure 3). When transferred to real datasets, the experiments results are rather weak.
>
> > **W1.1**: from UTKFace to FairFace (Table 1), the proposed approach (either with Laftr or CFair) does not achieve the best fairness in terms of equalized odds. For example, Laftr+FixMatch / CFair + FixMathc both achieve good accuracy and lower equalized odds than the proposed approaches. The group accuracy variance does decrease for the proposed approach, but why is that important compared to the fairness metrics? In addition, all the numbers have high standard deviations, are there any significance test done to show the superiority of some of the methods?
>
> **Answer to W1.1**:
> As explained in the general response, variance of group accuracy is very important. By looking at $\Delta_{odds}$ and $V_{acc}$ together, we can **avoid trivial fairness** (i.e., the model tends to predict a constant output, resulting in low equalized odds). In the UTKFace-FairFace experiment, although Laftr+FixMatch also achieves low equalized odds, such fairness is trivial and undesirable --- many examples from class 1 are classified to class 0, as shown in Figure 5 (SCR is using Laftr+FixMatch, and FCR is ours). By measuring group accuracy variance, we can easily recognize trivial fairness. Small group accuracy variance indicates that the model performs similarly for examples from different classes with different sensitive attributes.
>
> Compared with Laftr+FixMatch, **our method achieves similar equalize odds but reduces about 75% group accuracy variance**. As shown in Figure 5, our method (FCR) achieves similar accuracy for four groups where the fairness is non-trivial.
>
> We have also evaluated the Pareto frontiers of our method and baselines. Figure 8 in Appendix E.2 shows that our method achieves the best Pareto frontier, suggesting the superiority of our method in the trade-off between accuracy and fairness.
>
> Compared with baselines, our method has a smaller standard deviation. The high deviation of some of the baselines such as Laftr+DANN and Laftr+MMD are indeed what we observed which is also observed in the synthetic experiment. We suspect that it is because the domain adaptation method is less effective and stable for transferring both accuracy and fairness. Thanks for the suggestion, we will add a significance test in our next version.
>
>
> > **W1.2**: on the tabular dataset, the only results shown are in Figure 4. Other than LAFTR, no other baselines are shown, how do we know if the proposed methods work better than other baselines? Even just compared to LAFTR, from the figures (a) and (b) it is very hard to tell if the proposal made the results more fair (as most data points are almost in the same range, and there is no one-to-one correspondence). For Figure 4 (c), there is no significance test either, so it is unclear if the results are significant.
>
> **Answer to W1.2**:
> We have added the result of another baseline (Laftr+FixMatch) to Appendix E.4. Comparing Figure 4 (*a*) and (*b*) is indeed hard since we evaluate the model in all the US states. So, we plot Figure 4 (*c*) to see the improvement. From Figure 4 (*c*) and Figure 9 in Appendix E.4, we can see that Laftr+FixMatch increases the unfairness in more than half of the states, while our method decreases the unfairness in almost all the states, suggesting the superiority of our method in transferring fairness.
>
> We admitted in our paper that our method in this experiment is not as powerful as that for images. That is because self-training methods to tackle distribution shifts highly rely on the effectiveness of transformation functions (or data augmentation). For tabular data, transformation functions are very limited and under-explored. This is the limitation of our method, as discussed in appendix F. Finding good transformation functions for tabular data and using them to transfer accuracy and fairness are two orthogonal problems. We focus on the second one. So we believe our method will work better if future work can improve the transformation functions on tabular data.

---

> > ### Author Response · Authors · 2022-08-02
> > **$\newcommand{Rc}{\textcolor{blue}{15g2}}$ To Reviewer  $\Rc$ (2/2)**
> >
> >
> > > **W2**: The transformation/augmentation is an important piece in the overall framework, but it is not discussed in detail at all in this paper. The authors only briefly mentioned what transformations are excluded in Section 6.3. As the authors reuse existing augmentations designed for robustness (not necessarily for fairness), they should perform a more detailed study on how each of those transformations affect fairness, and if the effect is positive or negative.
> >
> > **Answer to W2**:
> > Thanks for the great suggestion. The transformation indeed plays an important role. Theoretically, as long as the transformation satisfies the intra-group expansion assumption (the basic requirement of which is that the transformation does not change the class or sensitive attribute), we can bound the error and unfairness as shown in Theorem 4.1. So, in our experiments, we select transformations based on this requirement. We would like to clarify that, by doing consistency regularization with those transformations, one can propagate labels from source to target so as to transfer accuracy. By using our proposed fair consistency regularization with a model that is fair in source, we can make different groups have similar accuracy gains in the target domain, so as to transfer fairness. In section 6.3, we investigate the role of transformation and show that when using weak transformations, the ability to transfer accuracy is limited but our method can still make the transferring process fair.
> >
> > We have added the experimental results of our method with 14 different transformations in Table 9 (Appendix E.3). Different transformations do have different effects on transferring fairness. We find that *Solarize*, *Color* and *TranslateX* increase the unfairness in the source domain the most, and *Contrast*, *Color* and *Solarize* have the highest unfairness in the target domain. Note that, it does not mean that these augmentations always lead to unfairness but that they are not suitable for our method. Our theory and algorithm are built upon the intra-group expansion assumption. Transformations like *Contrast*, *Color* and *Solarize* may change the sensitive attribute "race" and break this assumption. Thus, in our experiments (Table 1) we use all the transformations excluding them.
> >
> >
> > ---
> > Thanks again for reviewing our paper and asking good questions. We hope our answers have addressed all your questions and concerns. If so, we would greatly appreciate it if you could consider raising the score. We are happy to answer any follow-up questions.
> >
> > Authors

---

> > > ### Author Response · Authors · 2022-08-06
> > > **Any further comments?**
> > >
> > > $\newcommand{Rc}{\textcolor{blue}{15g2}}$
> > > Dear reviewer $\Rc$,
> > >
> > > Thank you again for your thoughtful review! Does our response address your questions? We would appreciate the opportunity to engage further if needed.
> > >
> > > Authors

---

> > > > ### Comment · Reviewer_15g2 · 2022-08-09
> > > > **Thanks for the response**
> > > >
> > > > I read the responses and want to thank the authors for the detailed responses to my question.
> > > >
> > > > I want to point out a few key points that should be better addressed in the main text of the paper:
> > > >
> > > > - If the authors want to propose a *new fairness metric* as the variance of group accuracy, then this decision should be better justified. Currently only line 84-86 describes this decision as "To avoid trivial fairness where a model with constant output has ∆odds = 0, we also evaluate the variance of group accuracy." I understand your reasoning about avoiding trivial fairness, but this is different from using this as one of the *main fairness metrics*. For example, what if a model outputs the same accuracy for every group, but has very unbalanced true positive rate (TPR) / false positive rate (FPR), yielding worse equalized odds?
> > > >
> > > > - In addition, the variance computes (acc-E(acc))^2 but the equalized odds metric computes the difference between TPR and FPR, so they could be of different scales and how do one compare one to another is another question. In Table 1, comparing CFair+FixMatch to the proposed approach (w/ CFair), is a decrease of 70% variance worth an increase of 40% on equalized odds?
> > > >
> > > > - Further, if the variance should indeed be considered as a fairness objective, then it's not a super fair comparison as existing methods only optimize for Equalized odds but not for this new fairness metric. In line 224-230, the authors added a group re-weighting process to encourage the model to focus on worse-performing groups, which could contribute to similar accuracies across groups (thus low variance of group accuracy) in the end. In this sense, should some of the DRO-based methods or group re-weighting methods also be compared with the proposed approach?
> > > >
> > > > Thanks the authors for adding the detailed comparison for each transformation. So it seems like 3 operations, Contrast, Color and Solarize are all excluded in Table 1? Please add those details (if space allows maybe even move table 9 to main text) in the experiments, as currently it only says "so we exclude transformations such as color jittering in RandAugment" (line 351).

---

> > > > > ### Author Response · Authors · 2022-08-09
> > > > > **Response to Reviewer 15g2's new comments**
> > > > >
> > > > >
> > > > > Thank you for the valuable points. We address them as follows, and will add them to the main paper.
> > > > >
> > > > > > Concerns about the variance of group accuracy.
> > > > >
> > > > > * We agree that we should introduce the variance of group accuracy at the beginning and emphasize the importance of it. We will revise the paper when we have more pages.
> > > > >
> > > > > * The variance of group accuracy can help to avoid trivial fairness. So, we suggest to first compare the variance of group accuracy and then compare the equalized odds. In Table 1, CFair+FixMatch has a very high variance of group accuracy comparing with ours, so the equalized odds is not comparable with ours since it is a trivial fairness. We will add this point to our main paper.
> > > > >
> > > > > * Our practical algorithm is based on our theoretical analysis of the fairness and accuracy under distribution shifts (one important contribution of this paper). The bounds of equalized odds (Theorem 4.1) and group accuracy variance (Appendix C) both suggests us to balance the consistency loss across groups as well as minimize the balanced consistency loss. To achieve this goal, we propose a dynamic re-weighting process. **We have compared with a static re-weighting process** in Table 3 which shows a better fairness of our method. DRO-based method might be another solution to balance the consistency loss but it needs many additional efforts to fit into our framework. We would like to emphasize that our algorithm is principle guided. Our contribution to algorithms is not only a dynamic re-weighting process. Even though there might be other solutions for balancing, we are the first to propose  balancing and minimizing the consistency loss to transfer accuracy and fairness. If we consider DRO-based methods or re-weighting methods outside our framework, we are not aware of any methods that can transfer fairness under domain shifts.
> > > > >
> > > > > > Questions about the transformation.
> > > > >
> > > > > * Yes, Contrast, Color and Solarize are all excluded in Table 1. We will add those details in the main paper. Thanks again for the great suggestion on adding Table 9.
> > > > >
> > > > >
> > > > > Thank you for your time in discussing with us. We hope our additional response have addressed your concerns. If so, could you please consider raising the score? We are happy to answer any follow-up questions.
> > > > >
> > > > > Authors

---

> > > > > > ### Comment · Reviewer_15g2 · 2022-08-09
> > > > > > **Response to the authors**
> > > > > >
> > > > > > Thanks for the clarification.
> > > > > >
> > > > > > I do not agree that the variance of group accuracy should be always prioritized before equalized odds. Also the usage of "trivial fairness" is rather ad-hoc and it only means the model outputs a constant resulting $\Delta_{odds}=0$, but clearly currently all the baselines have $\Delta_{odds}\neq 0$ so they do not result in "trivial fairness". Based on the current values of the variance and $\Delta_{odds}$ there should be a trade-off between those two metrics for each method, but I don't see a justification that one metric should be prioritized over another. As I mentioned above, what if a model outputs the same accuracy for every group, but has very unbalanced true positive rate (TPR) / false positive rate (FPR), yielding much worse equalized odds?
> > > > > >
> > > > > > I think currently too much emphasis has been put on the variance part as the main fairness metric, while this metric has not been well established (especially in the current main text there's only one sentence describing why the authors computed it).

---

> ### Author Response · Authors · 2022-08-08
> **May we know your thoughts on our response?**
>
> Dear Reviewer 15g2,
>
> Today is the last day of the rebuttal period. We would like to know if our response addresses your concerns and if there are any additional comments. We hope to use our last chance to respond to additional concerns. Your feedback is valuable to us. Thank you for your efforts and time!
>
> Authors

---

### Official Review · Reviewer_bbdE · 2022-07-04

**Rating:** 5
**Confidence:** 1
**Soundness:** 3 good
**Presentation:** 1 poor
**Contribution:** 2 fair

**Summary:**

In this work, the author analyses the problem of how model fairness is affected under different types of distribution shifts.
For this, they use equalized odd fairness metrics and propose several types of distribution shifts in the synthetic, real image, and real tabular datasets.
The authors also propose a "theory-minded algorithm" for transferring fairness with a fair consistency regularization.



**Questions:**

Other authors have clearly limited what relies upon to be possible or not to predict model performance degradation under distribution shift [1]
Do these limitations still apply when evaluating for transferring fairness?


**Limitations:**

Other authors have clearly limited what relies upon to be possible or not to predict model performance degradation under distribution shift [1]. This limitation is eluded by stating assumption #1

The abstract presents a "fine-grained analysis of how the fair model is affected under different types of distribution shift". There is no discussion about concept shift. This should be acknowledged.

**Strengths And Weaknesses:**

The paper deals with a highly relevant topic (fairness and distribution shift) that remains largely unexplored.

The authors select one measure of fairness equal opportunity odds, that is very relevant and widely used but without any social/redistributive justice justification.

The paper is in general hard to understand. Some of the mathematical formulations are not the traditional ones and they use overly complicated notations that difficult the reading of the paper.

The paper is also not self-contained and over relies on the appendix and external work that is not properly introduced.

Assumption #1 This assumption makes the problem solvable but in real cases the assumption does not hold, as one of the most common and challenging types of shift is concept shift, where the data generation process actually changes.
This happens more often with tabular data, and distinguishing covariate shift from concept shift is not feasible in many situations.
For the folktables data, one could arguably say that predicting US income by training in CA and evaluating has low fairness variations.
If the problem has a higher distribution shift e.g. by changing the prediction task to  "ACSTravelTime: predict whether an individual has a commute to work that is longer than 20 minutes" training in Hawaii and predicting in the rest of the states, assumption #1 will not hold. How would the proposed method work in this case? Would it flag that is not properly working? Will achieve better results than a baseline.



Minor remarks:
 - Line 15 "A synthetic dataset benchmark... is deployed" Is deployed the right verb here?
 - The figures are too small and the font size is considerably smaller than the text

[1] https://arxiv.org/abs/2201.04234

---

> ### Author Response · Authors · 2022-08-02
> **$\newcommand{Rb}{\textcolor{red}{bbdE}}$  To Reviewer $\Rb$ (1/2)**
>
> $\newcommand{Rb}{\textcolor{red}{bbdE}}$
>
> Thank you for your valuable feedback. Below, we address the questions and concerns in detail. Hope our answers can address your concerns.
>
> ---
> > **W1**: The authors select one measure of fairness equal opportunity odds, that is very relevant and widely used but without any social/redistributive justice justification.
>
> **Answer to W1**:
> Thanks for pointing this out. The selection of a fairness metric is a common problem in this area which usually requires expert knowledge for a particular application. Without expert knowledge, it is reasonable to use a general fairness metric that is applicable to many real applications. As the reviewer has mentioned, equalized odds is one of the most widely used metrics for classification tasks in existing papers. It encourages the model to achieve similar classification performance in different groups which has the social justice justification in many classification applications. That is why we use it in our paper.
>
> Additionally, our theoretical analysis can be extended to other fairness metrics that are based on the difference in group accuracy. For example, we also bound the variance of group accuracy (see Appendix C) with the variance of group consistency loss. This bound also suggests balancing the consistency loss as well as minimizing the balanced consistency loss, which is what our algorithm is doing.
>
> We will specify the motivation for using equalized odds and variance of group accuracy in our paper.
>
> ---
> > **W2**: The paper is in general hard to understand. Some of the mathematical formulations are not the traditional ones and they use overly complicated notations that difficult the reading of the paper.
>
> **Answer to W2**:
> We are sorry that the reviewer finds our notation confusing. Since we are dealing with the fairness under distribution shift problem where there are different domains, different classes, different groups, and different distributions, the notations are inevitably more complicated than those in papers that only care about fairness or distribution shifts. Following the notations used in [1] and [2], we've tried our best to simplify the notations. In fact, two other reviewers think our paper is clearly written. But we agree that we can future improve the clarity of the paper. We will add a notation table to help readers follow our notations more easily.
>
> > **W3**: The paper is also not self-contained and over relies on the appendix and external work that is not properly introduced.
>
> **Answer to W3**:
> We would like to respectfully argue that we believe the main paper is self-contained. The technique we used in this paper is indeed based on the recent progress of self-training in tackling distribution shifts [1, 2]. We extend their theory by taking fairness into consideration. Due to the page limitation, we defer the proofs, some discussions, and experimental details to the appendix. However, we made sure the main paper is self-contained. Line 42-49 introduces the main idea of [1] and [2], including the expansion assumption and why encouraging consistency can tackle domain shift. Line 50-57 introduces the difference between our work and [1, 2], and explains how we bound the unfairness. Line 58-66 introduces our practical algorithm and the major novelty. Sections 2,3 and 4 contain all the notations, definitions, assumptions, and theoretical results. Section 5 introduces the framework of our algorithm and the novel part of our algorithm - fair consistency regularization. We do not have a detailed introduction of Laftr and FixMatch, since they are two very famous works. Instead, we introduce their intuitions in the main paper and defer their loss functions to appendix D4 due to the page limitation. We will provide more details of these prior works if the page limit permits.
>
> Therefore, we believe our paper is self-contained. We would appreciate it if the reviewer could point out the missing piece. It would be very helpful for the improvement of our paper presentation.

---

> > ### Author Response · Authors · 2022-08-02
> > **$\newcommand{Rb}{\textcolor{red}{bbdE}}$  To Reviewer $\Rb$ (2/2)**
> >
> > > **W4**: Assumption 1 makes the problem solvable but in real cases, it does not hold, as one of the most common and challenging types of shift is concept shift, where the data generation process actually changes. This happens more often with tabular data, and distinguishing covariate shift from concept shift is not feasible in many situations. For the folktables data, if the problem has a higher distribution shift e.g. by changing the prediction task to "ACSTravelTime: predict whether an individual has a commute to work that is longer than 20 minutes" training in Hawaii and predicting in the rest of the states, assumption #1 will not hold. How would the proposed method work in this case? Would it flag that is not properly working? Will achieve better results than a baseline.
> >
> > **Answer to W4**:
> > Assumption 1 is very general and holds in many real cases, even in the ACSTravelTime problem. In Assumption 1, we assume that the underlying generative model is fixed, where $P_S(X|Y^{1:K}=y^{1:K})=P_T(X|Y^{1:K}=y^{1:K})$, while the marginal distribution of factors varies. Here, besides the label (e.g. commute time large than 20 minutes or not), the underlying factors also include other factors such as the location, the economic environment, the culture, and so on. All those factors together determine the observed data points. If one considers an incomplete set of latent factors (e.g. only commute time in an extreme case), one can say that the data generation process changes for different cities. However, as long as we consider a complete set of latent factors, the data generation process can be made to be the same in two domains. Therefore, Assumption 1 still holds for the ACSTravelTime problem. Additionally, the same data generation process assumption is also widely used in other papers that study distribution shifts such as [3].
> >
> > Assumption 1 does not necessarily exclude concept shift. Let's use $Y^1$ to denote the label and $Y^2,..,Y^K$ to denote other factors (we call them nuisance factors). Here, we do not consider sensitive attribute as a factor for simplicity. Since
> > $P(X|Y^1)=\sum_{Y^2,..,Y^K} P(X|Y^1,...,Y^K)P(Y^1,...,Y^K|Y^1)$, under Assumption 1, if the shift is caused by the marginal distribution shift of some nuisance factors (e.g. location), then $P_S(X|Y^1,...,Y^K)=P_T(X|Y^1,...,Y^K)$ and $P_S(Y^1,...,Y^K|Y^1)\neq P_T(Y^1,...,Y^K|Y^1)$, resulting in $P_S(X|Y^1)\neq P_T(X|Y^1)$. That's why we observe that the data from the two cities are so different even though they have the same label and share the same data generation process. $P_S(X|Y^1)\neq P_T(X|Y^1)$ can further be categorized into subpopulation shift and domain shift, as introduced in Section 3. Thus, Assumption 1 is very general. In fact, $P_S(Y^1|X)\neq P_T(Y^1|X)$ can also happen under Assumption 1, which we usually call it concept shift. We would like to clarify that since we also care about accuracy in both domains and it is not reasonable to use one model for two domains in this case if $P_S(Y^1|X)\neq P_T(Y^1|X)$, we also assume there is no concept shift. We will make this clear in our paper.
> >
> > How does our method work? We transfer fairness by encouraging the model to be fair under any nuisance factor values with the consistency loss. By doing so, when the marginal distribution of the nuisance factor changes, we can still maintain fairness. We apply transformations to X to simulate the change of the nuisance factor value. For images, we know many nuisance factors (e.g. light, angle) and we have effective transformation functions. However, it becomes hard for tabular data. The major challenge to transfer fairness on tabular data is to get transformation functions that can simulate, for example, the change of location or economic environment.
> >
> > ---
> >
> > > **Q1**: Other authors have clearly limited what relies upon to be possible or not to predict model performance degradation under distribution shift [1] Do these limitations still apply when evaluating for transferring fairness? This limitation is eluded by stating assumption #1
> >
> >
> > **Answer to Q1**:
> > As discussed above, Assumption 1 is very general and under which it is possible to transfer fairness. The true limitation relies on the difficulties of transformation functions that simulate the change of nuisance factor values. We have discussed it in appendix F.
> >
> > ---
> >
> > Thanks again for reviewing our paper and giving valuable questions. We hope our answers have addressed all the questions and concerns. If so, we would greatly appreciate it if you could consider raising the score. We are happy to answer any follow-up questions.
> >
> > Authors
> >
> > ---
> > Reference:
> >
> > [1] Wei, Colin, et al. "Theoretical analysis of self-training with deep networks on unlabeled data." ICLR 2021.
> >
> > [2] Cai, Tianle, et al. "A theory of label propagation for subpopulation shift." ICML 2021.
> >
> > [3] Wiles, Olivia, et al. "A fine-grained analysis on distribution shift." ICLR 2022.

---

> > > ### Author Response · Authors · 2022-08-06
> > > **Any further comments?**
> > >
> > > $\newcommand{Rb}{\textcolor{red}{bbdE}}$
> > >
> > > Dear reviewer $\Rb$,
> > >
> > > Thank you again for your thoughtful review! Does our response address your questions? We would appreciate the opportunity to engage further if needed.
> > >
> > > Authors

---

> ### Author Response · Authors · 2022-08-08
> **May we know your thoughts on our response?**
>
> Dear Reviewer bbdE,
>
> Today is the last day of the rebuttal period. We would like to know if our response addresses your concerns and if there are any additional comments. We hope to use our last chance to respond to additional concerns. Your feedback is valuable to us. Thank you for your efforts and time!
>
> Authors

---

### Official Review · Reviewer_FM6i · 2022-07-11

**Rating:** 6
**Confidence:** 3
**Soundness:** 3 good
**Presentation:** 3 good
**Contribution:** 3 good

**Summary:**

This paper aims to transfer fairness under distribution shift. Specifically, a comprehensive analysis of distribution shift is provided and then defines two types of distribution shift, named subpopulation shift and domain shift. Given the powerful self-training tool to solve the distribution shift problem, the authors first identify a sufficient condition for fairness transfer and then propose an algorithm with fair consistency regularization. Experiments on synthetic
and real datasets demonstrate the effectiveness in terms of fairness and accuracy under various distribution shifts.

Overall, this paper is generally good, in terms of technical novelty and evaluations.


**Questions:**

Please see Weaknesses.

**Ethics Review Area:**

["I don’t know"]

**Limitations:**

Yes

**Strengths And Weaknesses:**

Strengths

1.	This paper is well-written. The authors first provide a fine-grained analysis of fairness under distribution shift and identify the sufficient condition for fairness transfer. Motivated by the analysis of fairness, fair consistency regularization is introduced to transfer fairness under distribution shift.

2.	The idea of fair consistency regularization is simple yet effective with (partial) theoretical support.

Weakness

1.	There is a gap between theory (Theorem 4.1) and proposed algorithms. Theorem 4.1 implies that minimizing the worst-group consistency loss is beneficial to transfer fairness. However, the proposed regularization employs a balanced consistency loss in practice. It is still unclear, empirically or theoretically, why the worst-group consistency loss cannot work well in practice. Additionally, the dynamic weight seems to be important in the proposed regularization, what’s the rationale to design such dynamic weight?

2.	Marginal gain for the proposed algorithm. In figure 3 and table 1, the accuracy-fairness tradeoff performance seems to be marginal compared with baseline Laftr+FixMatch. Current results only provide one single accuracy and EO performance, it is hard to identify the superiority of the proposed method. A similar issue exists in Ablation study part. It would be better to plot the Pareto frontier of different methods with variable hyperparameters.

---

> ### Author Response · Authors · 2022-08-02
> **$\newcommand{Ra}{\textcolor{purple}{FM6i}}$     To Reviewer $\Ra$**
>
> $\newcommand{Ra}{\textcolor{purple}{FM6i}}$
>
> Thank you for the valuable feedback. We are particularly encouraged that you find our work novel, effective and well-written. Below, we address your questions in detail.
>
> ---
> > **W1**: There is a gap between theory (Theorem 4.1) and proposed algorithms. Theorem 4.1 implies that minimizing the worst-group consistency loss is beneficial to transfer fairness. However, the proposed regularization employs a balanced consistency loss in practice. It is still unclear, empirically or theoretically, why the worst-group consistency loss cannot work well in practice. Additionally, the dynamic weight seems to be important in the proposed regularization, what’s the rationale to design such dynamic weight?
>
> **Answer to W1**:
> From Theorem 4.1, we can see that the unfairness is upper bounded by the worst-group consistency loss, and the error is upper bounded by the all-groups consistency loss. Therefore, to train an accurate and fair model, we need to minimize all-groups consistency loss and the worst-group consistency loss at the same time. This requirement naturally leads to an algorithm that balances the consistency loss across groups as well as minimizes the balanced consistency loss. By doing this, we can find a model that has a small consistency loss for every group, so as to reduce both the error and unfairness.
>
>
> The rationale for designing dynamic weights is to achieve our goal of minimizing the balanced group consistency loss. One straightforward solution is to measure the consistency loss in each group and add them up with the same weight/coefficient. However, besides the loss value, there is another thing we need to consider. Since the consistency loss is measured with pseudolabels and we only consider confident examples when measuring it, the number of examples that used for measuring the consistency loss could be very different for different groups and we need to take it into consideration. For example, there are two groups that have the same consistency loss, but the consistency loss for one group is measured on many examples (i.e. the model is good at this group) while the consistency loss in the other group is measured on just a few examples (i.e. the model is not good at this group). In this case, we should give the latter group more attention by giving more weight to its consistency loss. To achieve this, we weigh each group inversely with the number of confident pseudolabels.
>
>
> Thank you for these two questions, we will revise our paper to make the logic and rationale clearer.
>
>
> ---
> > **W2**: Marginal gain for the proposed algorithm. In figure 3 and table 1, the accuracy-fairness tradeoff performance seems to be marginal compared with baseline Laftr+FixMatch. Current results only provide one single accuracy and EO performance, it is hard to identify the superiority of the proposed method. A similar issue exists in the Ablation study part. It would be better to plot the Pareto frontier of different methods with variable hyperparameters.
>
> **Answer to W2**:
> We respectfully do not agree with the marginal gain statement. Compared with Laftr+FixMatch, our method is much fairer. In Figure 3, compared with Laftr+FixMatch, our method reduces about 30% unfairness in the target domain under DShift and 55% under Hshift. In Table 1, compared with Laftr+FixMatch, **our method reduces about 75% unfairness (variance of group accuracy) in the target domain**. Please see the general response for why the variance of group accuracy is crucial.
>
> Thanks for the great suggestion of plotting the Pareto frontier. We've added the Pareto frontier of Laftr, Laftr+FixMatch, and ours on UTKFace-FairFace experiment in Appendix E.2. **Our method achieves the best Pareto frontier**, suggesting that our method outperforms others with a better trade-off between accuracy and fairness.
>
> ---
>
> We again thank you for reviewing our paper and providing suggestions. We hope our answers have addressed all your questions and concerns. Please let us know if there are more questions.
>
> Authors

---

> > ### Comment · Reviewer_FM6i · 2022-08-08
> > **Response**
> >
> > I appreciate the authors' detailed responses. My concerns have been well clarified/addressed. I will finalize my score after the discussion with other reviewers.

---

> > > ### Author Response · Authors · 2022-08-08
> > > **Thank you**
> > >
> > > Thank you for the positive feedback! We are glad that all the concerns are addressed. Thanks again for your time and support.
> > >
> > > Authors

---

### Author Response · Authors · 2022-08-02
**General Response**

$\newcommand{Ra}{\textcolor{purple}{FM6i}}$
$\newcommand{Rb}{\textcolor{red}{bbdE}}$
$\newcommand{Rc}{\textcolor{blue}{15g2}}$

We thank all the reviewers for their valuable feedback and insightful questions! We are particularly encouraged that they consider our research problem important ($\Ra$, $\Rb$, $\Rc$), our method novel ($\Ra$, $\Rc$), and the paper well-written ($\Ra$, $\Rc$). We address individual questions in separate responses. Here, we address one common question and outline the updates to the revised submission based on the reviews.

> **Q**: Why does the variance of group accuracy an important fairness metric?

**Answer to Q**:
In this paper, besides equalized odds, we also use the variance of group accuracy as the fairness metric. The variance of group accuracy is crucial due to the following reasons.
* First of all, a small group accuracy variance indicates that the model performs similarly in different groups (where one group is defined as a collection of examples that have the same class and sensitive attribute). Similar accuracy indicates that the model treats all the groups the same, which means the model is fair.
* Secondly, by looking at equalized odds and group accuracy variance, we can avoid trivial fairness. Fairness that is not based on accuracy is meaningless. In an extreme case where the model has a constant output, the equalized odds is zero, but it is trivial fairness. By measuring group accuracy variance, we can easily recognize trivial fairness.
* Therefore, in our paper, we evaluate the equalized odds and group accuracy variance together. Note that, besides equalized odds, we also bound the group accuracy variance (at the end of Appendix C). Both bounds suggest balance while minimizing the consistency loss which is the design principle of our algorithm.

**Paper Updates**:

Thanks for the suggestions from reviewers, we have added additional experimental results as follows. (We will move important results to the main paper when the page limit permits.)
* **[Figure 8 in Appendix E.2]** We plot the Pareto frontiers of our method and baselines. Our method gets better Pareto frontiers than baselines suggesting our method is more accurate and fair. ($\Ra$, $\Rc$)
* **[Table 9 in Appendix E.3]** We investigate the effect of using different transformations in our method by evaluating 14 different transformations. ($\Rc$)
* **[Figure 9 in Appendix E.4]** We compare our method with another baseline, Laftr+FixMatch, on the NewAdult experiment. Results show that our method outperforms it with a decrease of unfairness in almost all the US states. ($\Rc$)


We greatly appreciate the time and effort of all reviewers. We hope our responses and the paper updates can address all the questions and concerns. Please let us know if there are further questions.

Authors

---

### Meta-Review · Area_Chair_K9bL · 2022-08-24

**Recommendation:** Accept
**Confidence:** Less certain

**Metareview:**

The reviews are a bit divergent. While all the reviewers appreciate the clarity of the paper and the theory-inspired proposed algorithm, they raised some concerns e.g., on the assumption employed for obtaining a theoretical result, as well as on marginal (or worse) EO fairness performance in some cases. Although concerns on the fairness performance improvement in light of the employed metrics are still unresolved, many of the concerns are properly addressed, and with regard to the writing quality and insights, I believe that the paper is worth being published. Hence, I recommend the acceptance of this paper.

**Award:**

No

---

### Decision · Program_Chairs · 2022-09-14

Accept